# Adaptive Node Feature Selection For Graph Neural Networks

## Abstract

We propose an adaptive node feature selection approach for graph neural networks (GNNs) that identifies and removes unnecessary features during training. The ability to measure how features contribute to model output is key for interpreting decisions, reducing dimensionality, and even improving performance by eliminating unhelpful variables. However, graph-structured data introduces complex dependencies that may not be amenable to classical feature importance metrics. Inspired by this challenge, we present a model- and task-agnostic method that determines relevant features during training based on changes in validation performance upon permuting feature values. We theoretically motivate our intervention-based approach by characterizing how GNN performance depends on the relationships between node data and graph structure. Not only do we return feature importance scores once training concludes, we also track how relevance evolves as features are successively dropped. We can therefore monitor if features are eliminated effectively and also evaluate other metrics with this technique. Our empirical results verify the flexibility of our approach to different graph architectures as well as its adaptability to more challenging graph learning settings.

## 1 Introduction

Graphs provide powerful yet well-understood representations of complex data (Bronstein et al., 2017). Their rich modeling capabilities motivated the development of graph neural networks (GNNs) to exploit connectivity for predictive tasks (Wu et al., 2021). However, insufficient understanding of model decisions renders them untrustworthy for critical applications and potentially inefficient or suboptimal (Dong et al., 2022; Yuan et al., 2023; Wang & Ding, 2025; Chien et al., 2024). Deciphering how deep learning models extract information from data is challenging, particularly when data is equipped with complex interdependencies (Zhu et al., 2024). While some tools such as decision trees inherently provide model explanations, the most expressive tools are not directly interpretable and require explanation via heuristic-based metrics (Mandler & Weigand, 2024). As a prominent example, measuring feature importance is a fundamental technique for understanding how a model forms decisions (Wang et al., 2024). In particular, we are interested in determining how node features contribute to GNN outputs (Shao et al., 2024).

Beyond interpretability, identifying relevant attributes allows us to build models that are both economical and potent by eliminating unnecessary features (Li et al., 2018). Moreover, simplifying models can improve our understanding of complex real-world systems by reducing them to their most parsimonious representations (Georg et al., 2023). However, classical feature importance metrics do not account for an underlying graph structure and therefore may not be suitable for reducing nodal attributes (Chereda et al., 2024; Mahmoud et al., 2023). Additionally, past graph-based feature selection methods often involve assumptions about how graph structure contributes to learning, rendering these techniques problem-specific (Maurya et al., 2022; 2023; Zheng et al., 2025). To remove dependence on prior information, we may instead compute changes in model performance upon perturbing features to assess their contributions (Datta et al., 2016; Fisher et al., 2019). As these measurements require a trained model, feature selection using perturbation-based scores may require training multiple models, which can be costly for large-scale data or complicated architectures (Alkhoury et al., 2025). While some works train submodules to learn masks for identifying important features, these approaches can require learning additional parameters, undermining the

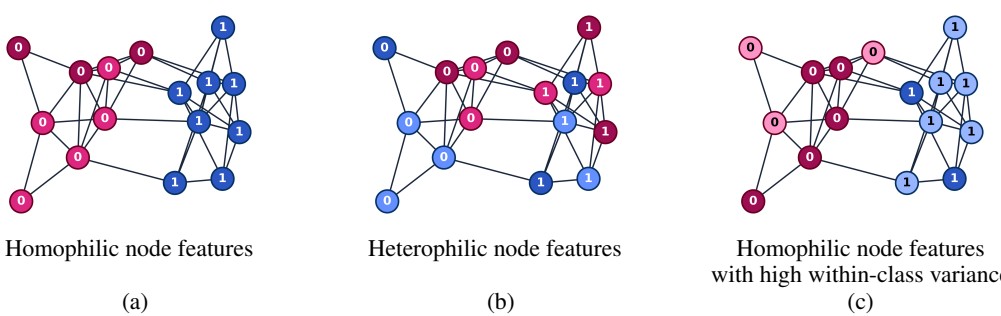

Figure 1: Example graphs for which graph structure can alter how node features affect node classification. Class labels are denoted by "0" or "1". Node features are represented by color, where red and blue indicate features from different distributions, and brightness indicates different magnitudes. (a) Edges directly imply similarity of node labels and features. (b) While most connected nodes belong to the same class, edges also tend to indicate distribution shifts in node features. (c) Both node labels and features are homophilic, but the high variance of node feature distributions may render classification more challenging.

goal of reducing dimensionality (Maurya et al., 2022; Acharya & Zhang, 2020; Lin et al., 2020; Zheng et al., 2020). A more in-depth overview of related works is shared in Appendix A.

Instead, we propose an *adaptive node feature selection algorithm* that measures *permutation-based feature importance during training* using GNN predictions. More specifically, we periodically permute the values of each node feature and measure changes in GNN performance on a validation dataset. Our scores are thus inherently tied to the predictive task, adapting to model learning and therefore allowing flexibility to GNN architecture, requiring no assumptions on graph data. Moreover, unlike graph-based feature selection works that use black-box models to learn importance values during training, we employ well-established permutation tests to quantify feature influence (Altmann et al., 2010; Yang et al., 2009; Breiman, 2001; Datta et al., 2016), allowing us to theoretically show how permutations reflect node feature influence. Our contributions are summarized below.

- We first characterize the effects of graph structure and node features on GNN performance, both theoretically and empirically. For the former, we show how connections influence the effect of node features on graph convolutional network (GCN) outputs. For the latter, we compare GNN accuracy under various perturbations to distinguish model dependence on graphs versus features.
- We propose an adaptive node feature selection approach that dynamically identifies which features are relevant to GNN performance via permutation-based importance scores. Because we measure these scores as training progresses, we can monitor how feature contributions change as the model evolves and variables are eliminated. We thus visualize importance scores during training to track model quality and verify that we indeed eliminate unhelpful attributes.
- We demonstrate that our algorithm rivals the performance of a GNN using all available node features in comparison with other node feature selection methods for multiple benchmark datasets. Furthermore, we show that our approach is flexible to model architecture and for various settings, such as homophilic or heterophilic node labels.

## 1.1 NOTATION

For any positive integer $N \in \mathbb{N}$, we define the notation $[N] := \{1, 2, \ldots, N\}$. For the vector $\mathbf{x} \in \mathbb{R}^N$, we index entries via $x_i$ for any $i \in [N]$, whereas for a matrix $\mathbf{X} \in \mathbb{R}^{N \times M}$, we index entries by $X_{ij}$, rows by $\mathbf{X}_{i,:}$, and columns by $\mathbf{X}_{:,j}$. We let boldfaced numbers $\mathbf{0}$ and $\mathbf{1}$ represent vectors or matrices of all zeros and ones, respectively. Furthermore, we have $\mathbf{I}$ as the identity matrix and $\mathbf{e}_i = \mathbf{I}_{:,i}$ as the $i$-th standard basis vector. For $\mathbf{0}, \mathbf{1}, \mathbf{I}$, and $\mathbf{e}_i$, we specify dimensions when it is unclear from context. The operator $\text{diag}(\mathbf{x}) \in \mathbb{R}^{N \times N}$ evaluated on a vector $\mathbf{x} \in \mathbb{R}^N$ returns a diagonal matrix with entries of $\mathbf{x}$ along the diagonal, while $\text{diag}(\mathbf{X}) \in \mathbb{R}^N$ for a square $\mathbf{X} \in \mathbb{R}^{N \times N}$ returns a vector of the diagonal entries of $\mathbf{X}$. We also let $\text{vec}(\mathbf{X}) \in \mathbb{R}^{NM}$ return the concatenation of columns in the matrix $\mathbf{X} \in \mathbb{R}^{N \times M}$. Moreover, let $\mathbb{I}(\cdot)$ denote the indicator function, where $\mathbb{I}(\mathcal{A}) = 1$ when its argument $\mathcal{A}$ is true and $\mathbb{I}(\mathcal{A}) = 0$ otherwise.

## 2 FEATURE IMPORTANCE FOR NODE CLASSIFICATION

We are interested in a semi-supervised node classification setup, where we have a graph $\mathcal{G} = (\mathcal{V}, \mathcal{E})$ consisting of a set of $N$ nodes $\mathcal{V}$ and a set of edges $\mathcal{E} \subseteq \mathcal{V} \times \mathcal{V}$ connecting pairs of nodes in $\mathcal{V}$. To use a graph in model training, we consider the adjacency matrix $\mathbf{A} \in \mathbb{R}_+^{N \times N}$, where $A_{ij} \neq 0$ if and only if the edge $(i, j) \in \mathcal{E}$ connects nodes $i$ and $j$, and $A_{ij} > 0$ denotes weight of the edge $(i, j)$. We can account for nodes with differing degrees $\mathbf{d} := \mathbf{A}\mathbf{1}$ by employing *normalized* adjacency matrices such as $\tilde{\mathbf{A}} := \tilde{\mathbf{D}}^{-1/2}(\mathbf{A} + \mathbf{I})\tilde{\mathbf{D}}^{-1/2}$ for $\tilde{\mathbf{D}} := \mathrm{diag}(\mathbf{d} + \mathbf{1})$ or a random-walk adjacency matrix $\tilde{\mathbf{A}}_{\mathrm{rw}} := \tilde{\mathbf{D}}^{-1}(\mathbf{A} + \mathbf{I})$ (Kipf & Welling, 2017). In addition to graph connections, each node is equipped with $M$ real-valued features, which we collect in the data matrix $\mathbf{X} \in \mathbb{R}^{N \times M}$. Furthermore, nodes are assigned labels $\mathbf{y} = [\mathbf{y}_{\mathrm{train}}^\top, \mathbf{y}_{\mathrm{val}}^\top, \mathbf{y}_{\mathrm{test}}^\top]^\top \in [C]^N$, of which we only observe a subset $[\mathbf{y}_{\mathrm{train}}^\top, \mathbf{y}_{\mathrm{val}}^\top] \in [C]^{N_{\mathrm{train}} + N_{\mathrm{val}}}$ for $N_{\mathrm{train}}, N_{\mathrm{val}} < N$. We also let $\mathbf{Y} \in \{0, 1\}^{N \times C}$ denote the one-hot matrix indicating the class of each node, along with $\mathbf{P} := \mathrm{diag}(\mathbf{p})$ for $\mathbf{p} := \mathbf{Y}^\top \mathbf{1} \in \mathbb{N}^C$, which contains the number of nodes in each class. We aim to predict the unknown labels $\mathbf{y}_{\mathrm{test}}$ by learning the parameters of a GNN $f(\cdot; \cdot, \boldsymbol{\Theta}) : \mathbb{R}^{N \times M} \to \mathbb{R}^{N \times H}$ that yields embeddings $\mathbf{Z} := f(\mathbf{X}; \mathbf{A}, \boldsymbol{\Theta})$ such that we may predict labels $\hat{\mathbf{y}} = g(\mathbf{Z})$ with some classifier $g : \mathbb{R}^{N \times H} \to [C]^N$.

Of particular relevance to us is how to identify which node features in $\mathbf{X}$ are important for predicting labels $\mathbf{y}$ while accounting for the graph structure $\mathbf{A}$ (Maurya et al., 2023; Chen et al., 2020). Some works apply traditional, graph-agnostic metrics to determine important features for a pre-trained GNN (Wang & Ding, 2025; Basaad et al., 2024; Chereda et al., 2024). However, the presence of edges used by the GNN can significantly alter which node features are relevant. For example, GCNs assume that edges directly indicate nodes that likely belong to the same class. Figure 1 illustrates how this assumption can alter how informative node features are. As GCNs are best suited to homophilic node features and labels as in Figure 1a, it is common to assess feature quality through its smoothness, that is, how similar feature values are between connected nodes (Zhu et al., 2024). However, even with homophilic node labels, a GCN applied to the graph in Figure 1b may not yield sufficiently separable node embeddings (Luan et al., 2024). Furthermore, if labels are homophilic and node features in different classes follow distinctly different distributions yet exhibit high variance, exemplified in Figure 1c, a graph-agnostic classifier may distinguish classes more easily than a GCN. Motivated by this consideration, we theoretically characterize how $\mathbf{A}$ and $\mathbf{X}$ influence GCN performance, which we then empirically verify on real-world graph data.

Recall that our goal is for our embeddings $\mathbf{Z} = f(\mathbf{X}; \mathbf{A}, \boldsymbol{\Theta})$ to be distinguishable across classes. A reasonable requirement for this task is that node embeddings exhibit sufficient separation across classes (Tenorio et al., 2025; Nt et al., 2021). However, we encounter at least two potential sources of error: noise in features $\mathbf{X}$ and in edges $\mathbf{A}$. For the former, we consider the *idealized* node features to be $\mathbf{X}^* := \mathbf{Y}\mathbf{P}^{-1}\mathbf{Y}^\top \mathbf{X}$, that is, the matrix closest to $\mathbf{X}$ whose rows are identical for nodes in the same class, or equivalently,

$$\mathbf{X}^* = \underset{\mathbf{X}^*}{\mathrm{argmin}} \|\mathbf{X}^* - \mathbf{X}\|_F^2 \quad \text{s.t.} \quad \mathbf{X}_{i,:}^* = \mathbf{X}_{j,:}^* \, \forall \, i, j \in [N] \text{ s.t. } y_i = y_j. \tag{1}$$

By (1), we obtain a notion of feature informativeness: Even if the rows of $\mathbf{X}^*$ are equivalent within classes, they may be very similar or even identical across classes, rendering classification effectively infeasible (Nt et al., 2021). Thus, we consider the features $\mathbf{X}$ to be informative enough if $\mathbf{X}^*$ contains distinct rows for different classes, indicating a sufficient shift in feature distributions across classes (Tenorio et al., 2025). Note that we define $\mathbf{X}^*$ as above for simplicity, representing the most straightforward relationship between informative features and labels $\mathbf{y}$; node classes containing distribution shifts can still yield informative predictions (Luan et al., 2024).

For the latter, it is well established that cross-class edges, that is, those connecting nodes of different classes, mar GCN performance (Zhu et al., 2020). Hence, we define the *idealized* graph $\mathbf{A}^*$ as having no cross-class edges, where

$$A_{ij}^* := \begin{Bmatrix} A_{ij}, & y_i = y_j \\ 0 & \text{otherwise} \end{Bmatrix} \quad \forall \, i, j \in [N]. \tag{2}$$

We then let $\boldsymbol{\Delta} := \mathbf{A} - \mathbf{A}^*$ collect all edges between nodes of different classes. We next characterize the performance of a GCN with respect to features $\mathbf{X}$, edges $\mathbf{A}$, and labels $\mathbf{y}$ by comparing our embeddings $\mathbf{Z}$ to the *idealized* ones $\mathbf{Z}^* := f(\mathbf{X}^*; \mathbf{A}^*, \boldsymbol{\Theta})$.

Table 1: Node classification accuracy for multiple datasets under various perturbations. The top performing method is **boldfaced**, and the secondmost underlined.

| Setting | Cora | CiteSeer | PubMed | Photo | Computers | Cornell | Texas | Wisconsin |
|---|---|---|---|---|---|---|---|---|
| $\text{GNN}(\mathbf{X}; \mathbf{A}, \boldsymbol{\Theta})$ | $\mathbf{85.83} \pm 0.46$ | $\mathbf{74.38} \pm 1.09$ | $\mathbf{88.85} \pm 0.42$ | $\mathbf{94.04} \pm 0.69$ | $\mathbf{90.58} \pm 0.79$ | $74.59 \pm 7.76$ | $\mathbf{82.70} \pm 4.05$ | $82.80 \pm 3.25$ |
| $\text{MLP}(\mathbf{X}; \boldsymbol{\Theta})$ | $74.21 \pm 1.40$ | $\underline{70.02} \pm 1.39$ | $\underline{88.65} \pm 0.41$ | $88.73 \pm 0.73$ | $81.63 \pm 0.75$ | $\mathbf{78.92} \pm 5.51$ | $\mathbf{82.70} \pm 2.16$ | $\mathbf{83.60} \pm 6.62$ |
| $\text{GNN}(\tilde{\mathbf{X}}; \mathbf{A}, \boldsymbol{\Theta})$ | $76.53 \pm 1.12$ | $63.34 \pm 1.44$ | $47.42 \pm 3.25$ | $66.76 \pm 8.58$ | $51.09 \pm 8.84$ | $43.78 \pm 8.95$ | $52.97 \pm 6.07$ | $46.80 \pm 6.76$ |
| $\text{GNN}(\mathbf{W}; \mathbf{A}, \boldsymbol{\Theta})$ | $\underline{82.92} \pm 1.54$ | $67.37 \pm 1.61$ | $76.23 \pm 0.53$ | $\underline{89.92} \pm 0.58$ | $\underline{85.49} \pm 0.44$ | $49.19 \pm 7.91$ | $55.68 \pm 2.76$ | $47.20 \pm 7.00$ |
| $\text{GNN}(\mathbf{X}; \tilde{\mathbf{A}}, \boldsymbol{\Theta})$ | $36.97 \pm 1.80$ | $35.13 \pm 2.41$ | $67.07 \pm 0.80$ | $30.52 \pm 5.77$ | $37.64 \pm 0.45$ | $\underline{67.57} \pm 7.83$ | $\underline{70.27} \pm 7.05$ | $\underline{78.40} \pm 6.37$ |

**Theorem 1** *Let $f : \mathbb{R}^{N \times M} \to \mathbb{R}^{N \times H}$ be a two-layer GCN*

$$f(\mathbf{X}; \mathbf{A}, \boldsymbol{\Theta}) = \sigma \left( \tilde{\mathbf{A}}_{\text{rw}} \sigma \left( \tilde{\mathbf{A}}_{\text{rw}} \mathbf{X} \boldsymbol{\Theta}^{(1)} \right) \boldsymbol{\Theta}^{(2)} \right) \quad (3)$$

*for a $\tau$-Lipschitz nonlinearity $\sigma$ and learnable weights $\boldsymbol{\Theta} = (\boldsymbol{\Theta}^{(1)}, \boldsymbol{\Theta}^{(2)})$ such that $\|\boldsymbol{\Theta}^{(\ell)}\|_2 \leq \omega$ for $\ell = 1, 2$. Then, with $\mathbf{Z}^* = f(\mathbf{X}^*; \mathbf{A}^*, \boldsymbol{\Theta})$ for $\mathbf{X}^*$ in (1), $\mathbf{A}^*$ in (2), and $\boldsymbol{\Delta} = \mathbf{A} - \mathbf{A}^*$, we have*

$$\|\mathbf{Z}^* - \mathbf{Z}\|_F \leq \tau^2 \omega^2 \left[ (1 + \sqrt{N}) \|\boldsymbol{\Delta}\|_F \|\mathbf{X}\|_F + \sum_{c=1}^{C} \sum_{i=1}^{N} \sum_{j=1}^{N} \left| \frac{Y_{ic} Y_{jc}}{p_c} - \frac{A_{ij}}{d_i + 1} \right| \cdot \|\mathbf{X}_{i,:} - \mathbf{X}_{j,:}\|_2 \right]. \quad (4)$$

The proof of Theorem 1 can be found in Appendix B. Thus, our GCN error bound depends on cross-class edges in $\mathbf{A}$ via the first term in (4) and the alignment of labels $\mathbf{y}$, edges in $\mathbf{A}$, and similarity of features in $\mathbf{X}$. First, we discuss when the presence of $\mathbf{A}$ necessitates stricter conditions on $\mathbf{X}$ for satisfactory GCN performance according to Theorem 1. While the result in (4) does not necessitate unweighted edges, the following discussion assumes $\mathbf{A} \in \{0, 1\}^{N \times N}$ for ease of interpretation. Unsurprisingly, GCNs require features $\mathbf{X}$ to be highly indicative of $\mathbf{y}$ if $\mathbf{A}$ is sparse or noisy. More specifically, for any node pair $i$ and $j$ in the same class $c$ that are not connected $(i, j) \notin \mathcal{E}$, we rely on similarity between node features $\|\mathbf{X}_{i,:} - \mathbf{X}_{j,:}\|_2$ to reduce the second term in (4). Thus, graph-agnostic feature importance metrics may be suitable for sparse $\mathbf{A}$. However, if $\mathbf{X}$ is separable across classes, that is, $\|\mathbf{X}_{i,:} - \mathbf{X}_{j,:}\|_2$ is higher when $Y_{ic} Y_{jc} = 0$, we incur greater error from cross-class connections $A_{ij} = 1$. In this setting, if we disregard $\mathbf{A}$ when selecting features, we may retain attributes that are highly separable with respect to classes, causing larger $\|\mathbf{X}_{i,:} - \mathbf{X}_{j,:}\|_2$ when $Y_{ic} Y_{jc} = 0$ and unknowingly introducing error due to cross-class edges in $\mathbf{A}$.

Conversely, the presence of $\mathbf{A}$ can also mitigate error due to noisy features $\mathbf{X}$. In particular, if $\mathbf{y}$ is sufficiently homophilic with respect to $\mathbf{A}$, that is, if $Y_{ic} Y_{jc} = A_{ij}$ holds for sufficiently many node pairs, then we can still achieve a low error via (4), even if $\mathbf{y}$ and $\mathbf{X}$ are unrelated. Moreover, the bound in (4) can be reduced when the variance in $\mathbf{X}$ is sufficiently dominated by class sizes $\mathbf{p}$ and node degrees $\mathbf{d}$, reflecting the intuitive fact that nodes with high degree $d_i$ belonging to a class of large size $p_c$ are easier to predict (Liu et al., 2023; Kang et al., 2022). Thus, Theorem 1 shows that measuring feature importance based solely on dependencies between $\mathbf{y}$ and $\mathbf{X}$ may not be sufficient for GNN feature selection (Zheng et al., 2024). More specifically, the bound in (4) reveals that certain compositions of features and edges may render a feature important or unimportant regardless of its relevance in the absence of the graph.

We next empirically verify the intuition from Theorem 1 by comparing GNN node classification accuracy on real-world benchmark datasets under various perturbations intended to remove dependencies among $\mathbf{X}$, $\mathbf{A}$, and $\mathbf{y}$. All simulation details can be found in Appendix D, which includes dataset details. To assess the joint influence of a graph and its features, we consider (i) $\text{GNN}(\mathbf{X}; \tilde{\mathbf{A}}, \boldsymbol{\Theta})$ using an Erdos-Renyi (ER) graph with the same number of edges as $\mathbf{A}$, (ii) $\text{MLP}(\mathbf{X}; \boldsymbol{\Theta})$, a multilayer perceptron (MLP) that considers no graph, (iii) $\text{GNN}(\mathbf{W}; \mathbf{A}, \boldsymbol{\Theta})$ using Gaussian noise $\mathbf{W} \sim \mathcal{N}(\mathbf{0}, \mathbf{I})$ as node features, and (iv) $\text{GNN}(\tilde{\mathbf{X}}; \mathbf{A}, \boldsymbol{\Theta})$, where $\tilde{\mathbf{X}}$ contains randomly permuted rows of $\mathbf{X}$. We train GCNs for Cora, Citeseer, and PubMed (Sen et al., 2008; Namata et al., 2012) and graph isomorphism networks (GINs) (Xu et al., 2019) for Photo and Computers (McAuley et al., 2015; Shchur et al., 2018), while for Cornell, Texas, and Wisconsin graphs with heterophilic labels (Pei et al., 2020), we consider topology adaptive GCNs (TAGCNs), which can aggregate features of nodes in multi-hop neighborhoods (Du et al., 2017).

Table 1 demonstrates both the importance of node features in these datasets along with their dependence on the associated graph. For the first five datasets, the respectable performance of $\text{MLP}(\mathbf{X}; \boldsymbol{\Theta})$ and $\text{GNN}(\mathbf{W}; \mathbf{A}, \boldsymbol{\Theta})$ demonstrates that both features $\mathbf{X}$ and graph structure $\mathbf{A}$ are semantically

relevant. We also observe particularly low accuracy for $\text{GNN}(\mathbf{X}; \tilde{\mathbf{A}}, \boldsymbol{\Theta})$, reflecting the error due to cross-class edges in (4), which is caused by the arbitrary connections in $\tilde{\mathbf{A}}$ despite the informativeness of $\mathbf{X}$. Furthermore, if $\mathbf{X}$ experiences significant shifts across classes, then applying permuted features via $\text{GNN}(\tilde{\mathbf{X}}; \mathbf{A}, \boldsymbol{\Theta})$ is expected to perform worse than $\text{GNN}(\mathbf{W}; \mathbf{A}, \boldsymbol{\Theta})$ for sparse $\mathbf{A}$ since summands in the second term of (4) may have large $\|\mathbf{X}_{i,:} - \mathbf{X}_{j,:}\|_2$ for $Y_{ic}Y_{jc} = 1$. Indeed, we find $\text{GNN}(\mathbf{W}; \mathbf{A}, \boldsymbol{\Theta})$ outperforms $\text{GNN}(\tilde{\mathbf{X}}; \mathbf{A}, \boldsymbol{\Theta})$ for all datasets in Table 1. We also corroborate known challenges of graph convolutions for data with heterophilic $\mathbf{y}$, as $\text{MLP}(\mathbf{X}; \boldsymbol{\Theta})$ rivals and can even outperform $\text{GNN}(\mathbf{X}; \mathbf{A}; \boldsymbol{\Theta})$ for Cornell, Texas, and Wisconsin. For these datasets, $\text{GNN}(\mathbf{X}; \tilde{\mathbf{A}}, \boldsymbol{\Theta})$ is significantly superior to $\text{GNN}(\tilde{\mathbf{X}}; \mathbf{A}, \boldsymbol{\Theta})$ and $\text{GNN}(\mathbf{W}; \mathbf{A}, \boldsymbol{\Theta})$, which reflects the difficulty of convolving node features that are both irrelevant to labels $\mathbf{y}$ and heterophilic on $\mathbf{A}$, as even random connections in $\tilde{\mathbf{A}}$ yield significantly higher accuracy.

## 3 PERMUTATION TESTS FOR NODE FEATURE IMPORTANCE

Inspired by Theorem 1 and Table 1, we propose *node feature permutation testing (NPT)* to measure feature importance via *permutation-based scores* (Altmann et al., 2010; Khan et al., 2025; Yang et al., 2009). In particular, let $\Pi$ be the set of permutations of $[N]$. Then, if $\tilde{\mathbf{X}}^{(m)}$ denotes $\mathbf{X}$ with values of feature $m$ reordered according to some random $\boldsymbol{\pi} \in \Pi$, we measure feature importance through permutation tests

$$\delta_m(\mathbf{y}, \mathbf{X}, \tilde{\mathbf{X}}^{(m)}) := \text{Acc}(\mathbf{y}, f(\mathbf{X}; \mathbf{A}, \boldsymbol{\Theta})) - \text{Acc}\big(\mathbf{y}, f(\tilde{\mathbf{X}}^{(m)}; \mathbf{A}, \boldsymbol{\Theta})\big), \tag{5}$$

where $\text{Acc}(\mathbf{y}, \mathbf{Z})$ measures the accuracy of embeddings $\hat{\mathbf{y}} = g(\mathbf{Z})$ for classifier $g$. With some abuse of notation, we let $\delta_m(\mathbf{y}_{\text{train}}, \mathbf{X}, \tilde{\mathbf{X}}^{(m)})$ denote the accuracy for the subset of nodes corresponding to observed training nodes, with analogous definitions for other subsets of nodes. Permutation tests are a classical approach to isolate the effects of a feature (Breiman, 2001; Toth, 2020; Altmann et al., 2010), and we next show that it can be particularly informative in the presence of $\mathbf{A}$. To this end, we validate that permuting columns of $\mathbf{X}$ indeed decouples node features from $\mathbf{y}$ and $\mathbf{A}$, which verifies that $\delta_m$ reflects feature influence for GCN predictions, supporting the results in Table 1.

**Theorem 2** *Consider* $\tilde{\mathbf{X}} \in \mathbb{R}^{N \times M}$ *such that* $\tilde{\mathbf{X}}_{i,:} = \mathbf{X}_{\pi(i),:}$ *for all* $i \in [N]$ *and some permutation* $\boldsymbol{\pi} \in \Pi$ *chosen uniformly at random. For the same GCN defined in* (3)*, let* $\tilde{\mathbf{Z}}^* := f(\tilde{\mathbf{X}}^*; \mathbf{A}^*, \boldsymbol{\Theta})$ *for* $\tilde{\mathbf{X}}^* := \mathbf{Y}\mathbf{P}^{-1}\mathbf{Y}^\top\tilde{\mathbf{X}}$*,* $\mathbf{A}^*$ *in* (2)*, and* $\boldsymbol{\Delta} = \mathbf{A} - \mathbf{A}^*$*. Furthermore, if* $\alpha := \max_{m \in [M]} \max_{k,\ell \in [N]} (X_{km} - X_{\ell m})^2$*, then with probability at least* $e^{-t^2/4}$*, we have that*

$$\|\tilde{\mathbf{Z}}^* - \tilde{\mathbf{Z}}\|_F \leq \tau^2 \omega^2 \Big[(1 + \sqrt{N})\|\boldsymbol{\Delta}\|_F \|\mathbf{X}\|_F + \sqrt{\gamma}\big\|\text{vec}\big(\mathbf{Y}\mathbf{P}^{-1}\mathbf{Y}^\top - \tilde{\mathbf{D}}^{-1}\mathbf{A}\big)\big\|_1\Big],$$

$$\text{where} \quad \gamma := \frac{2}{N-1}\left(\|\mathbf{X}\|_F^2 - \frac{1}{N}\|\mathbf{X}^\top\mathbf{1}\|_2^2\right) + \alpha t M \sqrt{N}. \tag{6}$$

We prove Theorem 2 in Appendix C. The bound in (6) reveals how comparing accuracy with and without permuting node features reveals the influence of $\mathbf{X}$. Because the heterophily of $\mathbf{y}$ encoded in $\|\text{vec}(\mathbf{Y}\mathbf{P}^{-1}\mathbf{Y}^\top - \tilde{\mathbf{D}}^{-1}\mathbf{A})\|_1$ can no longer be mitigated by node feature similarities as in the original bound (4), permuting features will likely worsen GCN performance if $\mathbf{X}$ is informative. However, a highly homophilic $\mathbf{y}$ can reduce the error bound in (6), implying less informative features $\mathbf{X}$, where a small bound in (6) relative to that of (4) implies low values of $\delta_m$. Similarly, we also observe that features with low variance will reduce $\gamma$ and therefore the error bound (6), as expected since features that exhibit smaller differences are likely to be less informative. Thus, Theorem 2 supports comparing GNN performance before and after permuting feature values to determine feature importance.

### 3.1 ADAPTIVE NODE FEATURE SELECTION

Given the value of permutation tests for node feature importance, we propose an adaptive feature selection method in Algorithm 1 to identify and remove unnecessary features during training. The matrix $\hat{\mathbf{X}}$ in Algorithm 1 denotes the pruned feature matrix with masked columns corresponding to $\mathbf{b} \in \{0, 1\}^M$, representing selected features. After the model $f$ has been trained for $T_{\text{burn}}$ epochs, we periodically compute the empirical average $\hat{\delta}_m$ of the NPT importance score $\delta_m(\mathbf{y}_{\text{val}}, \hat{\mathbf{X}}, \tilde{\mathbf{X}}^{(m)})$

---

**Algorithm 1:** Adaptive node feature selection via NPT.

---

**Input:** Step size $\lambda > 0$, $T_{\text{burn}}, T \in \mathbb{N}$, $K \in \mathbb{N}$, $r \in (0,1)$
1 Initialize $\hat{\mathbf{X}} = \mathbf{X}$, feature mask $\mathbf{b} \in \{0,1\}^M$, counter $t = 1$.
2 **while** *Stopping criteria not met* **do**
3     Gradient update: $\mathbf{\Theta} \leftarrow \mathbf{\Theta} - \lambda \nabla_{\mathbf{\Theta}} \mathcal{L}(\mathbf{y}_{\text{train}}, f(\hat{\mathbf{X}}; \mathbf{A}, \mathbf{\Theta}))$.
4     Update $t \leftarrow t + 1$.
5     **if** $t > \max(T, T_{\text{burn}})$ **then**
6         Reset $t \leftarrow 1$.
7         **for** $m \in \{\ell \mid b_\ell = 1, \ \ell \in [M]\}$ **do**
8             Initialize average score $\hat{\delta}_m = 0$.
9             **for** $k \in [K]$ **do**
10                 Sample random permutation $\boldsymbol{\pi} \sim \Pi$.
11                 Permute $\hat{\mathbf{X}}_{:,m}$ for $\tilde{\mathbf{X}}^{(m)}$ such that $\tilde{X}_{im}^{(m)} = \hat{X}_{\pi(i),m}$ and $\tilde{X}_{i\ell}^{(m)} = \hat{X}_{i\ell}$
12                     $\forall i \in [N], \ell \in [M] \backslash \{m\}$.
13                 Update $\hat{\delta}_m \leftarrow \hat{\delta}_m + \frac{1}{K} \delta_m(\mathbf{y}_{\text{val}}, \hat{\mathbf{X}}, \tilde{\mathbf{X}}^{(m)})$ via (5).
14             **end**
15         **end**
16         Compute $r$-quantile $\delta^{(r)}$ from $\{\hat{\delta}_m \mid b_m = 1, m \in [M]\}$.
17         **for** $m \in \{\ell \mid b_\ell = 1, \ \ell \in [M]\}$ **do**
18             **if** $\hat{\delta}_m < \delta^{(r)}$ **then**
19                 Prune unimportant feature $b_m \leftarrow 0$.
20             **end**
21         **end**
22     **end**
23 **end**

**Output:** Model $f(\cdot; \mathbf{A}; \mathbf{\Theta})$, pruned features $\hat{\mathbf{X}}$, mask $\mathbf{b}$, scores $\hat{\boldsymbol{\delta}}$

---

for every feature $m \in [M]$ over $K$ random permutations $\boldsymbol{\pi} \in \Pi$ (lines 7-15). We then keep the top $r$-th percentile of features based on $\hat{\boldsymbol{\delta}}$ by setting $b_m = 0$ for the remaining ones. We then continue training to update model parameters given the new subset of features. Algorithm 1 thus yields a single process to both train a GNN $f$ and successively prune unnecessary features. The most complex step of Algorithm 1 occurs at the first checkpoint when $t = T + 1$, where all $M$ features must be permuted $K$ times, resulting in $O(KNM)$. However, at $nT + 1$ for $n > 1$, we need only permute $r^n M < M$ features, so we may choose $r \in (0,1)$ with no cost to theoretical complexity.

In addition, advantages of Algorithm 1 include flexibility to graph data, architecture choice, and more. More specifically, since the metric $\delta_m$ is defined by changes in performance, we may replace accuracy Acc in (5) with any quality to which features ought to contribute, such as promoting fairness (Little et al., 2024; Navarro et al., 2024a;b). Thus, the model $f$ adapts to the learning task by the definition of $\delta_m$ without requiring prior assumptions on the graph, nor are we restricted to particular architectures (Maurya et al., 2022; 2023). Algorithm 1 is therefore amenable to various scenarios, including heteophilic labels $\mathbf{y}$ or features $\mathbf{X}$. Finally, while we espouse permutation tests due to our results in Theorems 1 and 2, line 14 may be computed using any feature importance score, as another may be particularly suited to the task given prior knowledge. However, many prior graph-based metrics do not account for model behavior (Mahmoud et al., 2023; Zheng et al., 2025), whereas $\delta_m$ explicitly aims to promote the accuracy of $f$, rendering it an appropriate general choice.

## 4 NUMERICAL EXPERIMENTS

We next evaluate our importance scores and algorithm based on node feature permutation tests. We consider the same datasets and architectures as in Table 1, with minimal details explained below. Dataset statistics, along with other dataset details, are included in Appendix D.

**Datasets.**

- **Citation networks:** Cora, Citeseer, and PubMed consist of papers as nodes, which are connected based on citations (Sen et al., 2008; Namata et al., 2012). The goal is to predict paper topic $\mathbf{y}$ from bag-of-words paper representations $\mathbf{X}$.

Table 2: Node classification accuracy for multiple datasets with feature selection. The top performing method is **boldfaced**, and the secondmost underlined.

| Method | Cora | CiteSeer | PubMed | Photo | Computers | Cornell | Texas | Wisconsin |
|---|---|---|---|---|---|---|---|---|
| All features | $85.83 \pm 0.46$ | $74.38 \pm 1.09$ | $88.85 \pm 0.42$ | $94.04 \pm 0.69$ | $90.58 \pm 0.79$ | $74.59 \pm 7.76$ | $82.70 \pm 4.05$ | $82.80 \pm 3.25$ |
| NPT | $\mathbf{79.19} \pm 2.45$ | $\mathbf{69.35} \pm 1.49$ | $\mathbf{87.11} \pm 0.75$ | $\mathbf{93.59} \pm 0.79$ | $\underline{90.09} \pm 0.51$ | $\mathbf{69.73} \pm 6.26$ | $\mathbf{72.97} \pm 9.21$ | $\mathbf{73.20} \pm 6.88$ |
| NPT-mask | $\underline{76.05} \pm 1.08$ | $\underline{68.12} \pm 1.69$ | $\underline{86.11} \pm 0.82$ | $93.48 \pm 0.59$ | $89.94 \pm 0.22$ | $63.24 \pm 4.39$ | $64.86 \pm 5.13$ | $72.40 \pm 7.31$ |
| TFI | $72.73 \pm 5.47$ | $65.77 \pm 2.04$ | $83.80 \pm 0.92$ | $93.02 \pm 0.68$ | $\underline{90.09} \pm 0.23$ | $61.62 \pm 7.13$ | $61.08 \pm 4.39$ | $52.40 \pm 3.44$ |
| MI | $66.83 \pm 3.68$ | $63.79 \pm 1.02$ | $85.96 \pm 1.00$ | $\underline{93.56} \pm 0.61$ | $\mathbf{90.33} \pm 0.32$ | $\underline{63.78} \pm 5.82$ | $\underline{65.41} \pm 7.33$ | $\underline{69.60} \pm 5.99$ |
| $h_{\text{attr}}$ | $39.96 \pm 1.00$ | $22.59 \pm 1.01$ | $78.85 \pm 0.21$ | $93.53 \pm 0.46$ | $\underline{90.09} \pm 0.46$ | $55.14 \pm 5.01$ | $58.38 \pm 5.01$ | $45.60 \pm 6.62$ |
| $h_{\text{Euc}}$ | $32.77 \pm 1.66$ | $22.62 \pm 1.03$ | $74.37 \pm 0.60$ | $\mathbf{93.59} \pm 0.50$ | $89.19 \pm 0.51$ | $52.43 \pm 4.05$ | $57.30 \pm 3.15$ | $44.00 \pm 7.48$ |
| $h_{\text{GE}}$ | $31.44 \pm 1.35$ | $22.47 \pm 1.01$ | $70.52 \pm 0.55$ | $93.41 \pm 0.46$ | $89.24 \pm 0.25$ | $52.43 \pm 4.05$ | $57.30 \pm 3.15$ | $44.00 \pm 7.48$ |
| Rnd. | $39.76 \pm 1.22$ | $34.39 \pm 4.07$ | $70.71 \pm 1.08$ | $91.99 \pm 0.49$ | $88.27 \pm 0.26$ | $56.65 \pm 4.69$ | $58.49 \pm 3.23$ | $57.04 \pm 4.49$ |

- **Co-purchase graphs:** Photo and Computers represent Amazon goods as nodes that are connected if frequently purchased together, with $\mathbf{y}$ as product category and $\mathbf{X}$ as word embeddings of product reviews (McAuley et al., 2015; Shchur et al., 2018).
- **Webpage graphs:** Cornell, Texas, and Wisconsin connect linked webpages of individuals in computer science departments across various universities (Pei et al., 2020). Labels $\mathbf{y}$ represent the role of individuals to be predicted from webpage word embeddings $\mathbf{X}$.

**Architectures.** Cora, Citeseer, and PubMed are trained with GCNs (Kipf & Welling, 2017), whereas we use TAGCNs for Cornell, Texas, and Wisconsin (Du et al., 2017). As for Photo and Computers, frequently co-purchased items likely indicate similar product categories, thus $\mathbf{y}$ is homophilic on $\mathbf{A}$. However, while reviews contain valuable keywords for prediction, positive and negative reviews may contain different words despite products belonging in the same category. Thus, since features $\mathbf{X}$ may exhibit both homophily and heterophily, we employ a GIN model, which can extract complex interactions of informative features (Xu et al., 2019).

**Metrics.** We compare our **NPT** scores to various alternative metrics for node feature importance. The full list can be found in Appendix D.3.

### 4.1 Node feature selection comparison

To validate our importance metric $\delta_m$ in (5), we train a GNN with the full set of features per dataset in Table 2, which we compare to GNNs trained with a subset of features selected based on **NPT** and other feature selection baselines described in Appendix D.3. For each method, we select the top $r\%$ of features ranked by the importance metric and retrain the GNN using only these features, with $r = 5\%$ for PubMed, Photo, and Computers and $r = 2\%$ otherwise. We observe that for all datasets, **NPT** achieves among the highest or the highest accuracy compared to other importance scores.

For the graphs with homophilic $\mathbf{y}$ (Cora, Citeseer, and PubMed), while both **NPT** and its masking variant **NPT-mask** outperform the rest, **NPT** is consistently superior. This aligns with our expectations from Theorems 1 and 2 that GCNs exhibit low error for features that are separable across classes, but permuting them will likely increase error more when permuted for graphs with homophilic labels. Indeed, Table 1 validates the informativeness of $\mathbf{X}$ for these three datasets, while Table 2 shows that permuting features is more effective at identifying relevant features for GCNs.

We similarly find that **NPT** performs best for the graphs Cornell, Texas, and Wisconsin with heterophilic labels, but we also witness worse performance for **NPT-mask**. Again, this follows our intuition from (4), which shows that for GCNs, reducing the variance for heterophilic features may actually decrease error, so masking features that are relevant to $\mathbf{y}$ by setting them to zero may underestimate their importance for settings of heterophily. On the contrary, the graph-agnostic **MI** performs well for Cornell, Texas, and Wisconsin in comparison with other metrics.

Finally, for Photo and Computers, we observe less differences in performance across all feature selection methods, even relative to **Rnd**, that is, randomly chosen features. Indeed, Table 1 indicated that graph structure is highly informative for node classification. Moreover, homophily metrics find much greater use for Photo and Computer than for the other datasets. As mentioned previously, review text containing keywords related to product category are likely to be not only homophilic but also correlated with labels $\mathbf{y}$, indicating the value of employing homophily-based scores for these two datasets. Similarly, for certain features, **MI** may be better able to identify review content

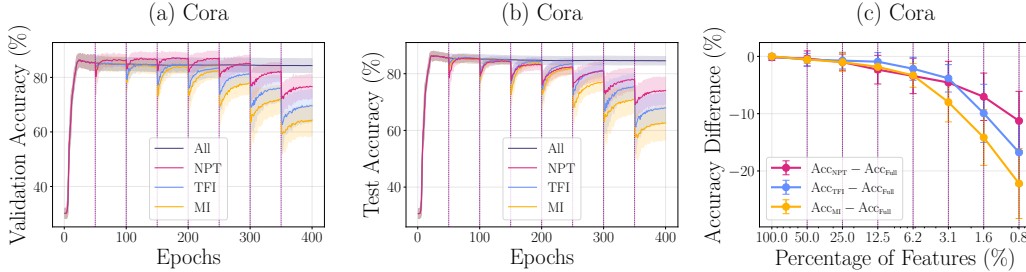

Figure 2: Node classification accuracy during training for a GCN and Cora using Algorithm 1 with different feature importance metrics. (a) Validation accuracy comparing a model trained using all features versus **NPT**, **TFI**, and **MI**. (b) Test accuracy comparing a model trained using all features versus **NPT**, **TFI**, and **MI**. (c) The difference in test accuracy between the full model and the model trained with Algorithm 1.

that is unrelated to $\mathbf{y}$ and $\mathbf{A}$ since it assumes no graph structure. We explore this context-specific information further for our adaptive approach.

## 4.2 ADAPTIVE NODE FEATURE SELECTION

Next, we assess our adaptive node feature selection approach. In particular, we apply Algorithm 1 to train GNNs while dropping less important features during training. To evaluate the tradeoff between maintaining performance and improving model efficiency, we evaluate accuracy in comparison with using the full dataset as the model is trained. At every $T = 50$ epochs, we drop 50% ($r = 0.5$) of features based on the scores $\delta_m$ in (5). Moreover, we also apply our algorithm with **TFI** and **MI** in place of $\delta_m$ to measure feature importance. Figure 2a,b depict accuracy during training to evaluate how models perform for the same number of features. For each checkpoint, we measure the difference between test accuracy using feature selection and the full dataset, shown in Figure 2c.

We present results for Cora in Figure 2, but we also include results for the remaining eight datasets mentioned in Appendix D.1. To evaluate on larger graphs, we also train GraphSAGE models (Hamilton et al., 2017) on the ArXiv citation network from the Open Graph Benchmark (Hu et al., 2020) with word embeddings as features and paper subjects as labels, for which we use $r = 0.4$. In Figure 2, we observe that **NPT** is better able to preserve accuracy than **MI** and even the GCN-specific **TFI** at low $r$. Furthermore, we observe smaller drops in accuracy for **NPT** as features are eliminated, as expected since our method adapts to GNN performance, allowing the model to focus on the importance of only the remaining features. We find similar comparisons of accuracy during training for the remaining datasets with **NPT** consistently demonstrating a competitive or superior ability to identify the most relevant features. Figures of accuracy during training analogous to Figure 2 can be found in Appendix F, while we provide a table including a subset of the results in Table 3. We find **NPT** effective for selecting important features during training, while **MI** is competitive for Cornell, Texas, and Wisconsin, similarly to Table 2. Moreover, we find **TFI** and **MI** to be effective importance metrics in our algorithm for Photo and Computers, which align with our intuition about these

Table 3: Node classification accuracy for multiple datasets with adaptive feature selection via Algorithm 1. The top performing method per ratio is **boldfaced**.

| % | Method | Cora | CiteSeer | PubMed | Photo | Computers | ArXiv | Cornell | Texas | Wisconsin |
|---|---|---|---|---|---|---|---|---|---|---|
| | NPT | **82.47** ± 1.68 | **71.82** ± 1.48 | **87.06** ± 0.89 | 83.54 ± 5.12 | 81.00 ± 2.42 | **40.84** ± 0.44 | 63.78 ± 3.67 | **72.43** ± 5.51 | 74.00 ± 6.07 |
| 6.25 | TFI | 81.40 ± 1.47 | 70.02 ± 2.04 | 84.25 ± 1.34 | **91.95** ± 1.10 | **84.47** ± 1.73 | − | 63.24 ± 8.48 | 61.08 ± 6.96 | 64.40 ± 6.86 |
| | MI | 78.23 ± 0.96 | 68.66 ± 1.98 | 86.48 ± 1.10 | 91.06 ± 1.06 | 83.82 ± 3.15 | 40.26 ± 0.18 | **66.49** ± 3.67 | 69.19 ± 3.67 | **78.00** ± 1.79 |
| | NPT | **81.88** ± 2.65 | **70.14** ± 1.50 | **86.51** ± 0.84 | 89.12 ± 2.29 | 86.92 ± 1.61 | **35.54** ± 0.63 | **67.57** ± 4.52 | 69.73 ± 8.44 | **69.60** ± 4.96 |
| 3.13 | TFI | 77.60 ± 1.13 | 68.30 ± 1.58 | 81.56 ± 1.66 | **93.05** ± 0.61 | **88.27** ± 0.74 | − | 63.24 ± 9.61 | 61.08 ± 5.57 | 58.00 ± 2.19 |
| | MI | 71.88 ± 1.48 | 65.11 ± 1.87 | 85.16 ± 1.01 | 92.39 ± 0.97 | 87.67 ± 1.65 | 35.30 ± 0.10 | 63.24 ± 4.05 | **70.81** ± 6.92 | 68.80 ± 5.74 |
| | NPT | **78.52** ± 2.17 | **69.08** ± 2.26 | **84.88** ± 0.69 | 88.71 ± 1.16 | 80.92 ± 4.33 | **30.58** ± 0.49 | **67.57** ± 3.82 | **70.27** ± 4.19 | **68.80** ± 5.46 |
| 1.56 | TFI | 71.73 ± 4.72 | 65.02 ± 1.82 | 79.38 ± 0.33 | 91.41 ± 0.94 | 86.65 ± 1.07 | − | 55.68 ± 6.30 | 60.54 ± 3.67 | 49.20 ± 5.46 |
| | MI | 63.51 ± 3.43 | 62.17 ± 0.49 | 83.41 ± 0.39 | **92.14** ± 0.55 | **86.87** ± 1.57 | 28.79 ± 1.32 | 60.54 ± 8.65 | 61.08 ± 10.76 | 66.00 ± 5.93 |

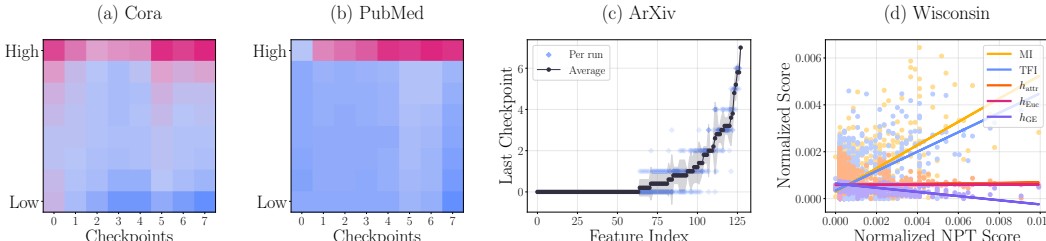

Figure 3: Analysis of feature importance scores obtained from Algorithm 1. (a) Heatmap of feature importance $\delta_m$ during training for a GCN trained on Cora (high $\delta_m$ is red and low $\delta_m$ is blue). (b) Heatmap of feature importance $\delta_m$ during training for a GCN trained on PubMed. (c) Last checkpoint each feature is kept before dropping for GraphSAGE trained on ArXiv. (d) Normalized importance scores per baseline versus normalized NPT scores for a TAGCN trained on Wisconsin.

datasets. Thus, with prior information, our algorithm can be further improved with an appropriate choice of metric, while permutation-based tests remain effective for general scenarios.

## 4.3 FEATURE IMPORTANCE ANALYSIS

We demonstrate our ability to dynamically track feature relevance during training, confirming that features can be appropriately dropped even before the model is fully trained. We exemplify periodically monitoring the scores $\delta_m$ in (5) for the Cora and PubMed datasets in Figure 3a,b. At each checkpoint, that is, every 50 epochs, we compute feature importance scores with **NPT**. When training is finished, we sort the features by the scores at the final checkpoint, corresponding to the fully trained model. We then fix the feature ordering based on their final scores and partition features into bins according to this ordering for each checkpoint. Thus, each row of each heatmap in Figure 3 represents the same set of features for the corresponding dataset, allowing us to track the average $\delta_m$ of each bin over time. For both datasets, we indeed identify relevant features as early as the first checkpoint, as the ranking of features is relatively consistent throughout training. This validates that with our adaptive approach, we can identify and preserve the relative importance of features even before full convergence, as the importance trends remain consistent over the course of training.

To illustrate the consistency of **NPT** feature selection, we also compute the average last checkpoint in which each ArXiv feature is kept before being dropped in Figure 3c. We find that the features of highest importance are consistently ranked high, while the least important features are always dropped early. For more concrete verification that **NPT** can identify importance in a controlled setting, Figure 12 in Appendix G visualizes importance scores using synthetic graph data, comparing scores obtained from **NPT**, **TFI**, **MI**, and **PT** (permutation testing via an MLP instead of a GNN).

Furthermore, we analyze the types of features deemed important by **NPT** across datasets, as well as verifying the generality of **NPT** as the metric in Algorithm 1. To this end, we plot normalized importance scores computed from baseline metrics versus **NPT** for Wisconsin in Figure 3d. We observe that **NPT** tends to rank Wisconsin features as more important with higher **MI** and lower homophily $h_{\text{GE}}$, as expected for data with heterophilic labels. To expand on this analysis, Table 4 lists the linear correlation between **NPT** scores and scores from each baseline for all datasets. In all cases, **NPT** attains its highest correlation with the metrics that performed best in Table 2. This result indicates two takeaways. First, **NPT** indeed identifies feature importance in based on relevant

Table 4: Pearson correlation coefficient between NPT feature importance $\delta_m$ and importance measured via other metrics. The top performing method is **boldfaced**, and the second best is underlined.

| Method | Cora | CiteSeer | PubMed | Photo | Computers | ArXiv | Cornell | Texas | Wisconsin |
|---|---|---|---|---|---|---|---|---|---|
| TFI | **0.6234** | **0.5032** | 0.2618 | 0.3098 | 0.3510 | — | 0.3659 | 0.3048 | 0.5225 |
| MI | 0.6178 | 0.4969 | **0.6026** | **0.5897** | **0.6380** | **0.5872** | **0.4314** | **0.4142** | **0.5658** |
| $h_{\text{attr}}$ | 0.1800 | 0.0373 | 0.2192 | 0.5260 | 0.6245 | 0.5656 | 0.0417 | -0.0093 | 0.0335 |
| $h_{\text{Euc}}$ | 0.0208 | 0.0266 | 0.1116 | 0.4152 | 0.2715 | 0.4523 | 0.0016 | -0.0171 | 0.0370 |
| $h_{\text{GE}}$ | -0.6479 | -0.5349 | -0.2285 | 0.1541 | 0.3072 | 0.0411 | -0.5712 | -0.5204 | -0.6635 |

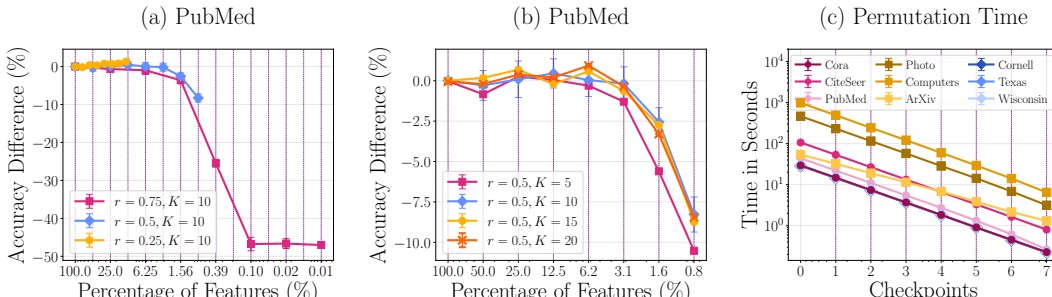

Figure 4: Evaluation of Algorithm 1 in various scenarios. (a) GCN performance on PubMed for fixed $K = 10$ and varying $r \in \{0.25, 0.5, 0.75\}$. (b) GCN performance on PubMed for fixed $r = 0.5$ and varying $K \in \{5, 10, 15, 20\}$. (c) Time to permute node features for each checkpoint of Algorithm 1, that is, for $t = nT$ for $n \in \mathbb{N}$, across multiple datasets.

data properties *without requiring prior information*. Second, because **NPT** detects relevant characteristics and attains competitive performance across datasets of various types, our approach is *a theoretically valid and empirically effective general choice for node feature selection*. Analogous plots of Figures 3c,d for the remaining datasets are in Appendix G.

### 4.4 Method performance analysis

We next demonstrate the performance of Algorithm 1 using **NPT** for PubMed while varying either $r$ or $K$, shown in Figure 4a,b. Further results can be found in Appendix H. As we drop more features via larger $r\%$, we naturally experience an increasing drop in accuracy. However, dropping features more slowly with $r = 0.25$ may improve performance, although we retain more features for the same number of training iterations. Moreover, we require a large enough $K$ to perform enough permutations for a statistically relevant result. In Figure 4b, increasing $K$ above our choice of 10 in previous simulations does not drastically change results, but lower $K$ can have negative effects on performance, as expected. For a statistical choice of $K$, see Proposition 1 in Appendix E. Finally, we measure the additional cost of permuting during training in Figure 4d. We observe the exact decay in permutation time as discussed in Section 3.1, where the cost of permutations is largest at the first checkpoint, but subsequent checkpoints decrease exponentially in duration. Moreover, as expected, graph size $N$ and the number of features $M$ control how costly computation will be, with dataset details listed in Appendix D.1.

## 5 Conclusion

In this work, we presented permutation tests for node feature importance. We verified the use of permutation-based importance scores for GCNs both theoretically and empirically. Furthermore, we presented an adaptive algorithm to eliminate features during training. We compared our permutation scores to other importance metrics for feature selection. We also demonstrated the effectiveness of our algorithm on multiple datasets, where we compared using permutation-based feature importance versus other metrics for adaptive feature selection. Our approach allows us to exploit a well-established statistical metric, but we also verified that it returns relevant information that is unique to GNNs for graph-structured data.

We also share limitations of this work that we hope inspire future directions. We require no assumptions on graph data, but performance-based metrics such as ours necessitate an appropriate selection of the GNN architecture. While a reasonable requirement, the interpretation of the importance scores may change depending on the model used. Furthermore, we demonstrated our approach only for node classification, but as $\delta_m$ can be employed to evaluate the effect of node features on any quantity, future work will see feature selection for link prediction and graph classification. Moreover, while permutation tests are typically found to be very effective (Khan et al., 2025), permuting features that are correlated may result in overestimated importance scores (Hooker et al., 2021). Thus, we plan to explore conditional permutation tests for the explainability of graph data. Finally, we expect that the performance of our algorithm can be improved further by adaptively eliminating features for which $\delta_m \leq 0$, which we explore in future work.

REPRODUCIBILITY STATEMENT

The model architectures, training hyperparameters, and experimental settings are detailed in Appendix D. Proofs of the theoretical results are given Appendix B and Appendix C. Dataset statistics are also reported in Appendix D.1. Finally, the source code and scripts are included in the supplementary materials, along with instructions to reproduce all experiments and results.

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

## A    RELATED WORK

Measuring variable importance is a fundamental task in several fields such as machine learning, statistics, and signal processing (Fisher et al., 2019; Mandler & Weigand, 2024). Classical techniques for classification tasks seek to identify correlations between features and labels to be predicted, such as their mutual information (Theng & Bhoyar, 2024). Simpler, interpretable models such as linear regression and decision trees can be used as surrogate models to explain sample or feature relevance (Ribeiro et al., 2016). To avoid training simple, albeit cheap, models, one of the most common approaches is to apply perturbations, where model inputs or parameters are perturbed and the change in output measured (Datta et al., 2016; Fisher et al., 2019; Covert et al., 2021). Seminal examples include feature occlusion (Feng et al., 2013; Lei et al., 2018), permutation (Altmann et al., 2010; Breiman, 2001; Datta et al., 2016), and Shapley values (Lundberg et al., 2018; Chen et al., 2019). Scores based on measuring model outcomes under perturbations may require training multiple models to be used for feature selection (Wang et al., 2024). Not only is this potentially infeasible computationally, but for optimizing models with nonconvex losses, differences in performance for models trained on perturbed data may be misleading.

For graph-structured data, a plethora of works seek to identify the contribution of nodes or edges to particular GNN predictions (Alkhoury et al., 2025; Akkas & Azad, 2024; Chen et al., 2024a; Huang et al., 2023). Among these, some works consider node feature relevance, albeit primarily as they pertain to structural importance (Fang et al., 2023; Chen et al., 2024b). Feature importance methods have been proposed specifically for graphs (Zheng et al., 2025), which often require assumptions about the type of graph data (Mahmoud et al., 2023; Shao et al., 2024). For example, as GCNs are a highly popular family of GNNs, the homophily of node features has been explored as relevance measurements (Zhu et al., 2024). The score proposed in (Zheng et al., 2025) computes the mutual information between labels and node features passed through a linear low-pass filter, implying relevance for a GCN. Authors considered all features informative, and their metric was used to identify which features ought to be trained with a GNN versus an MLP. Thus, they did not evaluate their metric for eliminating features to reduce model complexity or to remove unhelpful features. Conversely, several works aim to select node features during training, albeit without returning importance scores (Maurya et al., 2023; Jiang et al., 2023; Acharya & Zhang, 2020; Lin et al., 2020; Zheng et al., 2020). Moreover, these methods learn which features to eliminate via an auxiliary model, for which many tend to use uninterpretable models.

## B    PROOF OF THEOREM 1

The following proof is inspired by that of (Tenorio et al., 2025), which was itself motivated by (Nt et al., 2021) for evaluating GCN dependence on homophily.

By the definitions of $\mathbf{Z}^*$ and $\mathbf{Z}$, we have that $\mathbf{X}^* = \tilde{\mathbf{A}}_{\mathrm{rw}}^* \mathbf{X}^*$ and

$$\|\mathbf{Z}^* - \mathbf{Z}\|_F \leq \left\| \sigma_2 \left( \tilde{\mathbf{A}}_{\mathrm{rw}}^* \sigma_1 \left( \tilde{\mathbf{A}}_{\mathrm{rw}}^* \mathbf{X}^* \mathbf{\Theta}^{(1)} \right) \mathbf{\Theta}^{(2)} \right) - \sigma_2 \left( \tilde{\mathbf{A}}_{\mathrm{rw}} \sigma_1 \left( \tilde{\mathbf{A}}_{\mathrm{rw}} \mathbf{X} \mathbf{\Theta}^{(1)} \right) \mathbf{\Theta}^{(2)} \right) \right\|_F^2$$
$$\leq \tau\omega \left\| \tilde{\mathbf{A}}_{\mathrm{rw}}^* \sigma_1 \left( \tilde{\mathbf{A}}_{\mathrm{rw}}^* \mathbf{X}^* \mathbf{\Theta}^{(1)} \right) - \tilde{\mathbf{A}}_{\mathrm{rw}} \sigma_1 \left( \tilde{\mathbf{A}}_{\mathrm{rw}} \mathbf{X} \mathbf{\Theta}^{(1)} \right) \right\|_F^2,$$

with the latter inequality due to the $\tau$-Lipschitzness of $\sigma_2$ and the fact that $\|\mathbf{\Theta}^{(2)}\|_2 \leq \omega$. Then, we apply the triangle and Cauchy-Schwarz inequalities for

$$\|\mathbf{Z}^* - \mathbf{Z}\|_F \leq \tau\omega \left\| (\tilde{\mathbf{A}}_{\mathrm{rw}}^* - \tilde{\mathbf{A}}_{\mathrm{rw}}) \sigma_1(\mathbf{X}^* \mathbf{\Theta}^{(1)}) \right\|_F + \tau\omega \left\| \tilde{\mathbf{A}}_{\mathrm{rw}} (\sigma_1(\mathbf{X}^* \mathbf{\Theta}^{(1)}) - \sigma_1(\tilde{\mathbf{A}}_{\mathrm{rw}} \mathbf{X} \mathbf{\Theta}^{(1)})) \right\|_F$$
$$\leq \tau\omega \left\| \tilde{\mathbf{A}}_{\mathrm{rw}}^* - \tilde{\mathbf{A}}_{\mathrm{rw}} \right\|_F \left\| \sigma_1(\mathbf{X}^* \mathbf{\Theta}^{(1)}) \right\|_F + \tau\omega \left\| \tilde{\mathbf{A}}_{\mathrm{rw}} \left( \sigma_1(\mathbf{X}^* \mathbf{\Theta}^{(1)}) - \sigma_1(\tilde{\mathbf{A}}_{\mathrm{rw}} \mathbf{X} \mathbf{\Theta}^{(1)}) \right) \right\|_F.$$

Then, observing that $\|\tilde{\mathbf{A}}_{\mathrm{rw}}\|_2 = 1$ and exploiting the definitions of $\sigma_1$ and $\mathbf{\Theta}^{(1)}$, we have that

$$\|\mathbf{Z}^* - \mathbf{Z}\|_F \leq \tau^2\omega^2 \|\tilde{\mathbf{A}}_{\mathrm{rw}}^* - \tilde{\mathbf{A}}_{\mathrm{rw}}\|_F \|\mathbf{X}\|_F + \tau^2\omega^2 \|\mathbf{X}^* - \tilde{\mathbf{A}}_{\mathrm{rw}} \mathbf{X}\|_F. \tag{7}$$

Next, let $\tilde{\mathbf{D}}^* := \mathrm{diag}((\mathbf{A}^*+\mathbf{I})\mathbf{1})$, analogous to $\tilde{\mathbf{D}} = \mathrm{diag}(\mathbf{d}+\mathbf{1})$, and recall that $\tilde{\mathbf{A}}_{\mathrm{rw}} = \tilde{\mathbf{D}}^{-1}(\mathbf{A}+\mathbf{I})$ and $\tilde{\mathbf{A}}_{\mathrm{rw}}^* = (\tilde{\mathbf{D}}^*)^{-1}(\mathbf{A}^* + \mathbf{I})$. Then, we bound the adjacency matrix discrepancy as

$$\left\|\tilde{\mathbf{A}}_{\mathrm{rw}}^* - \tilde{\mathbf{A}}_{\mathrm{rw}}\right\|_F = \left\|\tilde{\mathbf{A}}_{\mathrm{rw}}^* - \tilde{\mathbf{D}}^{-1}(\mathbf{A}^* + \mathbf{I}) + \tilde{\mathbf{D}}^{-1}(\mathbf{A}^* + \mathbf{I}) - \tilde{\mathbf{D}}^{-1}(\mathbf{A} + \mathbf{I})\right\|_F$$

$$\leq \left\|\tilde{\mathbf{A}}_{\mathrm{rw}}^* - \tilde{\mathbf{D}}^{-1}(\mathbf{A}^* + \mathbf{I})\right\|_F + \left\|\tilde{\mathbf{D}}^{-1}(\mathbf{A}^* - \mathbf{A})\right\|_F$$

$$\leq \left\|(\mathbf{I} - \tilde{\mathbf{D}}^{-1}\tilde{\mathbf{D}}^*)\tilde{\mathbf{A}}_{\mathrm{rw}}^*\right\|_F + \left\|\tilde{\mathbf{D}}^{-1}\boldsymbol{\Delta}\right\|_F,$$

where $\tilde{\mathbf{D}}^{-1}(\mathbf{A}^* + \mathbf{I}) = \tilde{\mathbf{D}}^{-1}\tilde{\mathbf{D}}^*\tilde{\mathbf{A}}_{\mathrm{rw}}^*$. Recalling that $\|\tilde{\mathbf{A}}_{\mathrm{rw}}^*\|_2 = 1$, we then have

$$\left\|\tilde{\mathbf{A}}_{\mathrm{rw}}^* - \tilde{\mathbf{A}}_{\mathrm{rw}}\right\|_F \leq \left\|\tilde{\mathbf{D}}^{-1}(\tilde{\mathbf{D}} - \tilde{\mathbf{D}}^*)\right\|_F + \|\boldsymbol{\Delta}\|_F$$

$$\leq \|\mathbf{D} - \mathbf{D}^*\|_F + \|\boldsymbol{\Delta}\|_F$$

$$= \|\mathrm{diag}(\boldsymbol{\Delta}\mathbf{1})\|_F + \|\boldsymbol{\Delta}\|_F$$

$$\leq \|\boldsymbol{\Delta}\mathbf{1}\|_2 + \|\boldsymbol{\Delta}\|_F$$

$$\leq \sqrt{N}\|\boldsymbol{\Delta}\|_F + \|\boldsymbol{\Delta}\|_F, \tag{8}$$

which accounts for the first term in (7). For the second, we apply the definitions $\tilde{\mathbf{X}}^* = \mathbf{Y}\mathbf{P}^{-1}\mathbf{Y}^\top\mathbf{X}$ and $\tilde{\mathbf{A}}_{\mathrm{rw}} = \tilde{\mathbf{D}}^{-1}(\mathbf{A} + \mathbf{I})$ for

$$\|\mathbf{X}_{i,:}^* - [\tilde{\mathbf{A}}_{\mathrm{rw}}\mathbf{X}]_{i,:}\|_2 = \left\|\frac{1}{p_{y_i}}\sum_{j=1}^N Y_{j,y_i}\mathbf{X}_{j,:} - \frac{1}{d_i+1}\sum_{j=1}^N [\mathbf{I} + \mathbf{A}]_{ij}\mathbf{X}_{j,:}\right\|_2$$

$$= \left\|\frac{1}{p_{y_i}}\sum_{j=1}^N \mathbf{Y}_{j,:}\mathbf{Y}_{i,:}^\top\mathbf{X}_{j,:} - \frac{1}{d_i+1}\mathbf{X}_{i,:} - \frac{1}{d_i+1}\sum_{j=1}^N A_{ij}\mathbf{X}_{j,:}\right\|_2,$$

where the inner product $\mathbf{Y}_{j,:}\mathbf{Y}_{i,:}^\top = 1$ if and only if $y_i = y_j$. Then, we have that

$$\|\mathbf{X}_{i,:}^* - [\tilde{\mathbf{A}}_{\mathrm{rw}}\mathbf{X}]_{i,:}\|_2 = \left\|\frac{1}{p_{y_i}}\sum_{j=1}^N \mathbf{Y}_{j,:}\mathbf{Y}_{i,:}^\top\mathbf{X}_{j,:} - \mathbf{X}_{i,:} + \left(\frac{d_i}{d_i+1}\right)\mathbf{X}_{i,:} - \frac{1}{d_i+1}\sum_{j=1}^N A_{ij}\mathbf{X}_{j,:}\right\|_2$$

$$= \left\|\frac{1}{p_{y_i}}\sum_{j=1}^N \mathbf{Y}_{j,:}\mathbf{Y}_{i,:}^\top(\mathbf{X}_{j,:} - \mathbf{X}_{i,:}) + \frac{1}{d_i+1}\sum_{j=1}^N A_{ij}(\mathbf{X}_{i,:} - \mathbf{X}_{j,:})\right\|_2$$

$$= \left\|\sum_{j=1}^N \left(\frac{\mathbf{Y}_{j,:}\mathbf{Y}_{i,:}^\top}{p_{y_i}} - \frac{A_{ij}}{d_i+1}\right)(\mathbf{X}_{i,:} - \mathbf{X}_{j,:})\right\|_2$$

$$= \left\|\sum_{j=1}^N \left[\mathbf{Y}\mathbf{P}^{-1}\mathbf{Y}^\top - \tilde{\mathbf{D}}^{-1}\mathbf{A}\right]_{ij}(\mathbf{X}_{i,:} - \mathbf{X}_{j,:})\right\|_2,$$

which then leads to

$$\|\mathbf{X}^* - \tilde{\mathbf{A}}_{\mathrm{rw}}\mathbf{X}\|_F = \left[\sum_{i=1}^N \left\|\sum_{j=1}^N \left[\mathbf{Y}\mathbf{P}^{-1}\mathbf{Y}^\top - \tilde{\mathbf{D}}^{-1}\mathbf{A}\right]_{ij}(\mathbf{X}_{i,:} - \mathbf{X}_{j,:})\right\|_2^2\right]^{1/2}$$

$$\leq \sum_{i=1}^N\sum_{j=1}^N \left\|\left[\mathbf{Y}\mathbf{P}^{-1}\mathbf{Y}^\top - \tilde{\mathbf{D}}^{-1}\mathbf{A}\right]_{ij}(\mathbf{X}_{i,:} - \mathbf{X}_{j,:})\right\|_2$$

$$= \sum_{c=1}^C\sum_{i=1}^N\sum_{j=1}^N \mathbb{I}(y_i = c)\mathbb{I}(y_j = c)\cdot\left\|\left[\mathbf{Y}\mathbf{P}^{-1}\mathbf{Y}^\top - \tilde{\mathbf{D}}^{-1}\mathbf{A}\right]_{ij}(\mathbf{X}_{i,:} - \mathbf{X}_{j,:})\right\|_2$$

$$= \sum_{c=1}^C\sum_{i=1}^N\sum_{j=1}^N \left\|\left(\frac{Y_{ic}Y_{jc}}{p_c} - \frac{A_{ij}}{d_i+1}\right)(\mathbf{X}_{i,:} - \mathbf{X}_{j,:})\right\|_2. \tag{9}$$

Substituting (8) and (9) into the inequality (7) yields the result in (4), as desired. ∎

## C    PROOF OF THEOREM 2

Observe that we may repeat the steps of the proof of Theorem 1 in Appendix B to obtain

$$\|\tilde{\mathbf{Z}}^* - \tilde{\mathbf{Z}}\|_F \leq \tau^2\omega^2\left[(1 + \sqrt{N})\|\boldsymbol{\Delta}\|_F\|\mathbf{X}\|_F + \sum_{i=1}^N\sum_{j=1}^N\left|\frac{\mathbf{Y}_{i,:}\mathbf{Y}_{j,:}^\top}{p_{y_i}} - \frac{A_{ij}}{d_i + 1}\right|\cdot\|\tilde{\mathbf{X}}_{i,:} - \tilde{\mathbf{X}}_{j,:}\|_2\right]. \tag{10}$$

Thus, for the remainder of the proof, we need only obtain a bound for $\|\tilde{\mathbf{X}}_{i,:} - \tilde{\mathbf{X}}_{j,:}\|_2$. We proceed with bounding $(\tilde{X}_{im} - \tilde{X}_{jm})^2$ for any $m \in [M]$. For a given permutation $\boldsymbol{\pi} \in \Pi$ sampled uniformly at random, we define the function

$$\psi(\boldsymbol{\pi}) := \left(X_{\pi(i),m} - X_{\pi(j),m}\right)^2 \tag{11}$$

with expected value

$$\begin{aligned}
\mathbb{E}[\psi(\boldsymbol{\pi})] &= \mathbb{E}\left[\left(X_{\pi(i),m} - X_{\pi(j),m}\right)^2\right] \\
&= \mathbb{E}\left[X_{\pi(i),m}^2\right] - 2\mathbb{E}\left[X_{\pi(i),m}X_{\pi(j),m}\right] + \mathbb{E}\left[X_{\pi(j),m}^2\right].
\end{aligned} \tag{12}$$

Then, for any $i, j \in [N]$ such that $i \neq j$ and $m \in [M]$,

$$\mathbb{E}[X_{\pi(i),m}^2] = \frac{1}{N!}\sum_{\boldsymbol{\pi}\in\Pi}X_{\pi(i),m}^2 \;=\; \frac{1}{N!}\sum_{j=1}^N\sum_{\boldsymbol{\pi}\in\Pi}X_{jm}^2\mathbb{I}(\pi(i) = j) \;=\; \frac{1}{N}\sum_{j=1}^N X_{jm}^2 \;=\; \frac{1}{N}\|\mathbf{X}_{:,m}\|_2$$

and

$$\begin{aligned}
\mathbb{E}[X_{\pi(i),m}X_{\pi(j),m}] &= \frac{1}{N!}\sum_{\boldsymbol{\pi}\in\Pi}X_{\pi(i),m}X_{\pi(j),m} \\
&= \frac{1}{N!}\sum_{k=1}^N\sum_{\ell=1}^N\sum_{\boldsymbol{\pi}\in\Pi}X_{km}X_{\ell m}\mathbb{I}(\pi(i) = k)\mathbb{I}(\pi(j) = \ell) \\
&= \frac{1}{N!}\sum_{k\neq\ell}\sum_{\boldsymbol{\pi}\in\Pi}X_{km}X_{\ell m}(N - 2)! \\
&= \frac{1}{N(N-1)}\sum_{k=1}^N\sum_{\ell=1}^N X_{km}X_{\ell m} - \frac{1}{N(N-1)}\sum_{k=1}^N X_{km}^2 \\
&= \frac{1}{N(N-1)}\left((\mathbf{1}^\top\mathbf{X}_{:,m})^2 - \|\mathbf{X}_{:,m}\|_2^2\right),
\end{aligned}$$

which we substitute into (12) for

$$\mathbb{E}[\psi(\boldsymbol{\pi})] = \frac{2}{N-1}\left(\|\mathbf{X}_{:,m}\|_2^2 - \frac{1}{N}(\mathbf{1}^\top\mathbf{X}_{:,m})^2\right). \tag{13}$$

Then, we define the Doob martingale $\{Q_k\}_{k=0}^N$ such that $Q_0 = \mathbb{E}[\psi(\boldsymbol{\pi})]$ and

$$Q_k = \mathbb{E}[\psi(\boldsymbol{\pi}) \mid \pi(1), \ldots, \pi(k-1)] \quad \forall k = 1, \ldots, N,$$

thus $Q_N = \psi(\boldsymbol{\pi})$. Additionally, we have that

$$\mathbb{E}[\psi(\boldsymbol{\pi}) \mid \pi(1), \ldots, \pi(k-1)] = \sum_{\ell=k}^N\frac{1}{N-k+1}\mathbb{E}[\psi((k\ell)\boldsymbol{\pi}) \mid \pi(1), \ldots, \pi(k)],$$

where $\psi((k\ell)\boldsymbol{\pi})$ denotes $\psi$ given $\boldsymbol{\pi}$ with elements $k$ and $\ell$ swapped. Then, we bound the following differences

$$
\begin{aligned}
|Q_k - Q_{k-1}| &= \Big|\mathbb{E}[\psi(\boldsymbol{\pi})\,|\,\pi(1),\ldots,\pi(k)] - \mathbb{E}[\psi(\boldsymbol{\pi})\,|\,\pi(1),\ldots,\pi(k-1)]\Big| \\
&= \left|\mathbb{E}[\psi(\boldsymbol{\pi})\,|\,\pi(1),\ldots,\pi(k)] - \sum_{\ell=k}^{N}\frac{1}{N-k+1}\mathbb{E}[\psi((k\ell)\boldsymbol{\pi})\,|\,\pi(1),\ldots,\pi(k)]\right| \\
&= \left|\frac{1}{N-k+1}\sum_{\ell=k}^{N}\mathbb{E}[\psi(\boldsymbol{\pi})\,|\,\pi(1),\ldots,\pi(k)] - \mathbb{E}[\psi((k\ell)\boldsymbol{\pi})\,|\,\pi(1),\ldots,\pi(k)]\right| \\
&\leq \frac{1}{N-k+1}\sum_{\ell=k}^{N}\Big|\mathbb{E}[\psi(\boldsymbol{\pi})-\psi((k\ell)\boldsymbol{\pi})\,|\,\pi(1),\ldots,\pi(k)]\Big|. \qquad (14)
\end{aligned}
$$

Then, with $\tilde{\mathbf{X}}_{:,m}^{(k\ell)}$ representing $\tilde{\mathbf{X}}_{:,m}$ with elements $k$ and $\ell$ swapped, we have

$$
|\psi(\boldsymbol{\pi})-\psi((k\ell)\boldsymbol{\pi})| = \left|\big(\tilde{X}_{im}-\tilde{X}_{jm}\big)^2 - \big(\tilde{X}_{im}^{(k\ell)}-\tilde{X}_{jm}^{(k\ell)}\big)^2\right|,
$$

which is zero for $i=k, j=\ell$ or $i\neq k, j\neq \ell$, but for $k=i, j\neq \ell$, we instead have

$$
\begin{aligned}
|\psi(\boldsymbol{\pi})-\psi((k\ell)\boldsymbol{\pi})| &= \left|\big(\tilde{X}_{im}-\tilde{X}_{\ell m}\big)\big(\tilde{X}_{im}+\tilde{X}_{\ell m}-2\tilde{X}_{jm}\big)\right| \\
&= \big|\tilde{X}_{im}-\tilde{X}_{\ell m}\big|\cdot\big|\tilde{X}_{im}+\tilde{X}_{\ell m}-2\tilde{X}_{jm}\big| \\
&\leq \big|\tilde{X}_{im}-\tilde{X}_{\ell m}\big|\cdot\Big(\big|\tilde{X}_{im}-\tilde{X}_{jm}\big|+\big|\tilde{X}_{\ell m}-\tilde{X}_{jm}\big|\Big) \\
&\leq 2\alpha
\end{aligned}
$$

by our assumption that $\max_{m\in[M]}\max_{k,\ell\in[N]}(X_{km}-X_{\ell m})^2 = \alpha$. With (11), (13), and (14), we apply the Azuma–Hoeffding inequality (Azuma, 1967; Hoeffding, 1963) for

$$
\mathbb{P}\left[\psi(\boldsymbol{\pi}) - \mathbb{E}[\psi(\boldsymbol{\pi})] \geq \eta\right] \leq \exp\left(-\frac{\eta^2}{2N\alpha^2}\right),
$$

and with $t := \eta/(\alpha\sqrt{N})$, we obtain

$$
\big(X_{\pi(i),m}-X_{\pi(j),m}\big)^2 \leq \frac{2}{N-1}\left(\|\mathbf{X}_{:,m}\|_2^2 - \frac{1}{N}\mathbf{1}^\top\mathbf{X}_{:,m}\right) + \alpha t\sqrt{N}
$$

with probability at least $e^{-t^2/4}$, which yields

$$
\begin{aligned}
\|\tilde{\mathbf{X}}_{i,:}-\tilde{\mathbf{X}}_{j,:}\|_2 &= \sqrt{\left[\sum_{m=1}^{M}\frac{2}{N-1}\left(\|\mathbf{X}_{:,m}\|_2^2 - \frac{1}{N}(\mathbf{1}^\top\mathbf{X}_{:,m})^2\right) + \alpha t\sqrt{N}\right]} \\
&\leq \sqrt{\frac{2}{N-1}\left(\|\mathbf{X}\|_F^2 - \frac{1}{N}\|\mathbf{X}^\top\mathbf{1}\|_2^2\right) + M\alpha t\sqrt{N}},
\end{aligned}
$$

which we substitute into (10) for the error bound in (6), as desired. ∎

# D   EXPERIMENTAL DETAILS

This section provides further details regarding the simulations in this work, including the datasets employed.

## D.1   DATASET DETAILS

We share the statistics of the datasets used in our experiments in Table 5. Information about dataset context, that is, the interpretation of nodes, features, edges, and labels is provided in Section 4.

| Dataset | #Nodes | #Edges | #Feats | #Classes |
|---|---|---|---|---|
| Cora | 2,708 | 10,556 | 1,433 | 7 |
| CiteSeer | 3,327 | 9,104 | 3,703 | 6 |
| PubMed | 19,717 | 88,648 | 500 | 3 |
| Photo | 7,650 | 119,043 | 745 | 8 |
| Computers | 13,752 | 245,778 | 767 | 10 |
| ogbn-arxiv | 169,343 | 1,166,243 | 128 | 40 |
| Cornell | 183 | 298 | 1,703 | 5 |
| Texas | 183 | 325 | 1,703 | 5 |
| Wisconsin | 251 | 515 | 1,703 | 5 |

Table 5: Statistics of the benchmark datasets used in our experiments, including the number of nodes, edges, input features per node, and class labels.

### D.2 TABLE 1 SIMULATION DETAILS

We elaborate on training details for the results in Table 1. All results are averaged over five runs, except for the results in Figures 2 and 5-8, which are averaged over twenty runs. In each run, we randomly split the nodes into 70% training, 10% validation, and 20% test, and we report the test accuracy corresponding to the epoch with the highest validation accuracy. The train/validation/test masks are re-sampled independently for each run. For Cora, CiteSeer, and PubMed, we use a 2-layer GCN; for Amazon Computers and Photo we use a 2-layer GIN; for Cornell, Texas, and Wisconsin we use a 2-layer TAGCN; and for the MLP baseline we use a 2-layer MLP, all with 512 hidden units. Models on Computers and Photo are trained for 800 epochs, while the remaining datasets are trained for 400 epochs. We use the Adam optimizer with a learning rate of 0.01 and weight decay of $5 \times 10^{-4}$.

### D.3 TABLE 2 BASELINES AND SIMULATION DETAILS

We next describe our process for the results in Table 2. Our evaluation follows a two-stage pipeline: in the first stage, we train a model following the setup in Table 1 and compute feature importance scores using the validation set. We then select the top $r\%$ of features according to each FS method ($r = 2\%$ for all datasets except PubMed, Photo, and Computers, where $r = 5\%$ due to their smaller feature dimension). In the second stage, we retrain the model using only these selected features, with the same architecture and training configuration as in the first stage, and report the test accuracy.

As for baselines, we evaluate several feature selection (FS) methods listed below.

- **NPT:** Our **node feature permutation tests (NPTs)** for feature importance ranks features based on the drop in validation accuracy upon permuting each feature.
- **NPT-mask:** We introduce a variant of **NPT** where, rather than permuting a feature to remove its effect, we instead mask its values, that is, set all of its values to zero.
- **MI:** We measure the mutual information (MI) between each feature and the node labels.
- **TFI:** The **Topological Feature Informativeness (TFI)** metric was introduced in (Zheng et al., 2025) to measure feature importance prior to training to be applicable for GCNs.
- **Feature homophily:** The homophily-based metrics, $h_{\text{attr}}$ (Yang et al., 2021), $h_{\text{Euc}}$ (Chen et al., 2023), and $h_{\text{GE}}$ (Jin et al., 2022), score features according to different measures of homophily, that is, measuring the smoothness of each node feature according to different distance metrics.
- **Rnd.:** Our **random (Rnd.)** selection baseline, where we select features uniformly at random to be retained or removed.

### D.4 TABLE 3 AND FIGURE 2 SIMULATION DETAILS

For adaptive node feature selection, we set $r = 0.5$ in Algorithm 1 for all datasets, dropping half of the features at each step of the feature importance calculation, except for ArXiv, where we use $r = 0.4$ due to its relatively small feature dimension (128). The burn-in period $T_{\text{burn}}$ and interval period $T$ are fixed to 50 for all datasets, except for Computers, Photos, and arxiv, where $T_{\text{burn}}, T = 100$. The model is trained for 400 epochs on all datasets and 800 epochs on Computers, Photos, and arxiv.

Test accuracy for each feature percentage is reported based on the epoch with the highest validation accuracy: within each $T$ interval, we identify the epoch that achieves the best validation accuracy and use its corresponding test accuracy. This procedure is applied consistently across all feature selection methods. The architecture and optimizer settings follow the configuration described in Section D.2.

# E CHOICE OF NUMBER OF FEATURE PERMUTATIONS

Let $\{\tilde{\mathbf{x}}^{(k)}\}_{k=1}^{K}$ denote $K \in \mathbb{N}$ independent permutations of the vector $\mathbf{x} \in \mathbb{R}^N$, where $\tilde{x}_i = x_{\pi^{(k)}(i)}$ for every $i \in [N]$ for i.i.d. $\pi^{(k)} \in \Pi$. We seek to sample a large enough $K$ such that the empirical expected value $\frac{1}{K}\sum_{k=1}^{K} \tilde{\mathbf{x}}^{(k)}$ approximates the true expected value $\mathbb{E}[\tilde{\mathbf{x}}^{(k)}] = \mu\mathbf{1}$ for any $k \in [K]$, where $\mu := \frac{1}{N}\mathbf{1}^\top\mathbf{x}$. This will indicate that the empirical distribution of feature permutations approximates the true distribution. To this end, we consider the following result.

**Proposition 1** *For the vector $\mathbf{x} \in \mathbb{R}^N$, we define $\{\tilde{\mathbf{x}}^{(k)}\}_{k=1}^{K}$ such that $\tilde{x}_i = x_{\pi^{(k)}(i)}$ for every $k \in [K]$ and $i \in [N]$, where $\pi^{(k)} \in \Pi$ denote i.i.d. permutations of $[N]$. Then, with $x_{\max} := \max_i |x_i|$, we have that*

$$\mathbb{P}\left[\left\|\frac{1}{K}\sum_{k=1}^{K}\tilde{\mathbf{x}}^{(k)} - \mathbb{E}[\tilde{\mathbf{x}}^{(k)}]\right\|_2^2 \le \frac{t\sqrt{K-1}}{2K}\right] \ge 1 - 2\exp\left\{-\frac{Kt^2}{4N^2(x_{\max}^2 - \mu^2)^2}\right\}. \tag{15}$$

Thus, we may choose $K$ in Algorithm 1 such that our feature permutations are similar enough to the true distribution of random feature permutations, where we determine a satisfactory similarity via choice of $t$. The proof of Proposition 1 is as follows.

**Proof of Proposition 1.** First, given that $\mathbb{E}[\tilde{\mathbf{x}}^{(k)}] = \mu\mathbf{1}$, we have that

$$\left\|\frac{1}{K}\sum_{k=1}^{K}\tilde{\mathbf{x}}^{(k)} - \mathbb{E}[\tilde{\mathbf{x}}^{(k)}]\right\|_2^2 = \sum_{i=1}^{N}\left(\frac{1}{K}\sum_{k=1}^{K}(\tilde{x}_i^{(k)} - \mu)\right)^2$$

$$= \frac{1}{K^2}\sum_{i=1}^{N}\sum_{k=1}^{K}\sum_{\ell=1}^{K}(\tilde{x}_i^{(k)} - \mu)(\tilde{x}_i^{(\ell)} - \mu)$$

$$= \frac{1}{K^2}\sum_{k=1}^{K}\sum_{\ell=1}^{K}(\tilde{\mathbf{x}}^{(k)} - \mu\mathbf{1})^\top(\tilde{\mathbf{x}}^{(\ell)} - \mu\mathbf{1}).$$

Since $\pi^{(k)}$ are independently sampled uniformly at random from $\Pi$, for each $k, \ell \in [K]$ such that $k \ne \ell$, there exists some permutation $\rho^{(j)} \in \Pi$ such that $(\tilde{\mathbf{x}}^{(k)} - \mu\mathbf{1})^\top(\tilde{\mathbf{x}}^{(\ell)} - \mu\mathbf{1}) = (\mathbf{x} - \mu\mathbf{1})^\top(\hat{\mathbf{x}}^{(j)} - \mu\mathbf{1})$, where $\hat{\mathbf{x}}^{(j)}$ denotes the permutation of $\mathbf{x}$ by $\rho^{(j)}$ for every $j \in [J]$ with $J := K(K-1)/2$. Thus, our next step is to apply Hoeffding's inequality. To this end, first observe that the $j$-th inner product $(\mathbf{x} - \mu\mathbf{1})^\top(\hat{\mathbf{x}}^{(j)} - \mu\mathbf{1})$ denotes an independent random variable bounded between $-N|x_{\max}^2 - \mu^2|$ and $N|x_{\max}^2 - \mu^2|$. Then, for any $t_0 > 0$, Hoeffding's inequality states that

$$\mathbb{P}\left[\left|\sum_{j=1}^{J}(\mathbf{x} - \mu\mathbf{1})^\top(\hat{\mathbf{x}}^{(j)} - \mu\mathbf{1})\right| > t_0\right] \le 2\exp\left\{-\frac{t_0^2}{2JN^2(x_{\max}^2 - \mu^2)^2}\right\}.$$

Recalling that $J = \frac{K(K-1)}{2}$, we then let $t = \frac{t_0\sqrt{K-1}}{J}$ for

$$\mathbb{P}\left[\left\|\frac{1}{K}\sum_{k=1}^{K}\tilde{\mathbf{x}}^{(k)} - \mu\mathbf{1}\right\|_2^2 > t\frac{\sqrt{K-1}}{2K}\right] \le 2\exp\left\{-\frac{Kt^2}{4N^2(x_{\max}^2 - \mu^2)^2}\right\},$$

as desired. ∎

## F  ADDITIONAL PLOTS ON ADAPTIVE NODE FEATURE SELECTION

We present additional plots analogous to those in Figure 2 measuring the accuracy of GNNs trained using either all features available in a dataset versus using our adaptive Algorithm 1. We compare our approach using our proposed permutation-based node feature importance scores, and we also evaluate using **TFI** and **MI** as importance metrics to rank feature relevance in Algorithm 1. Figure 5 presents results for the datasets with homophilic labels Cora, CiteSeer, and PubMed; Figure 6 the datasets with heterophilic labels Cornell, Texas, and Wisconsin; and Figure 7 the larger-scale datasets Photo, Computers, and ArXiv. To further verify our results, we repeat our experiments on homophilic datasets Cora, CiteSeer, and PubMed using their official dataset splits into training, validation, and testing in Figure 8.

## G  ADDITIONAL PLOTS ON FEATURE IMPORTANCE ANALYSIS

This section includes Figure 9, which contains additional plots analogous to Figure 3c. For each feature, we measure the average last checkpoint of Algorithm 1 in which a feature is kept before being dropped for all datasets. In addition, Figure 10 plots the average last checkpoint for each feature rank, that is, the most frequent last checkpoint assigned to each feature, analogous to Figure 3d. We also include additional plots analogous to Figure 11 for all datasets.

Finally, we plot in Figure 12 the feature importance for synthetic graph data measured by **NPT**, **TFI**, **MI**, and **PT**, which is analogous to **NPT** but uses an MLP instead of a GNN to compute feature importance. More specifically, we generate five independent trials of graphs of $N = 500$ nodes and $M = 50$ features. We assign nodes to one of $C = 2$ classes. We vary the relationships between features, labels, and graph structure as follows.

- **Graph structure $\mathbf{A} \leftrightarrow$ labels $\mathbf{y}$**: When the graph and labels are independent ($\mathbf{A} \perp\!\!\!\perp \mathbf{y}$), we generate an Erdos-Renyi graph with edge probability 0.1. Otherwise, when ($\mathbf{A} \not\!\perp\!\!\!\perp \mathbf{y}$), we sample the graph from a stochastic block-model whose communities correspond to classes, where within-class edges are sampled with probability 0.1 and across-class edges with 0.05.
- **Graph structure $\mathbf{A} \leftrightarrow$ node features $\mathbf{X}$**: When the graph and features are independent ($\mathbf{A} \perp\!\!\!\perp \mathbf{X}$), we sample node features as Gaussian white noise $\mathbf{X}_0 \sim \mathcal{N}(\mathbf{0}, \sigma \mathbf{I})$ for $\sigma = 3$. Otherwise, when $\mathbf{A} \not\!\perp\!\!\!\perp \mathbf{X}$, we obtain the eigendecomposition of $\mathbf{A} = \mathbf{V}\mathbf{\Lambda}\mathbf{V}^\top$ and generate bandlimited graph signals as $\mathbf{X}_0 = \mathbf{V}_{:,\mathcal{B}}\mathbf{W}$ for $\mathbf{W} \sim \mathcal{N}(\mathbf{0}, \sigma \mathbf{I})$, where $\mathcal{B}$ denotes the indices of graph frequencies in $\mathrm{diag}(\mathbf{\Lambda})$ that are below $\lambda_{\max} = 0.5|\max_i \Lambda_{ii}|$.
- **Labels $\mathbf{y} \leftrightarrow$ node features $\mathbf{X}$**: When the graph and labels are independent ($\mathbf{y} \perp\!\!\!\perp \mathbf{X}$), we further process node features by sampling $\mathbf{B}_0 \sim \mathcal{N}(\mathbf{0}, \mathbf{I})$ for $\mathbf{B}_0 \in \mathbb{R}^{C \times 5}$. We normalize the columns of $\mathbf{B}_0$ to sum to zero and rescale for $\mathbf{B} = 5\mathrm{diag}^{-1}(|\mathbf{B}_0|\mathbf{1})\mathbf{B}_0$, where $|\mathbf{B}_0|$ denotes the element-wise absolute value of entries of $\mathbf{B}_0$. Finally, we update relevant entries of $\mathbf{X}$ as $\mathbf{X}_{:,m} = [\mathbf{X}_0]_{:,m} + [\mathbf{YB}]_{:,m}$ for $m \in [5]$. Otherwise, when $\mathbf{y} \not\!\perp\!\!\!\perp \mathbf{X}$, we simply let $\mathbf{X} = \mathbf{X}_0$.

## H  ADDITIONAL PLOTS ON MODEL PERFORMANCE ANALYSIS

We include additional plots in Figure 13 on hyperparameter tuning for Cora, CiteSeer, and PubMed, which correspond to Figure 4a,b. In particular, we fix $r = 0.5$ and vary $K \in \{5, 10, 15, 20\}$ in the top row, whereas for the bottom row, we fix $K = 10$ and vary $r \in \{0.25, 0.5, 0.75\}$.

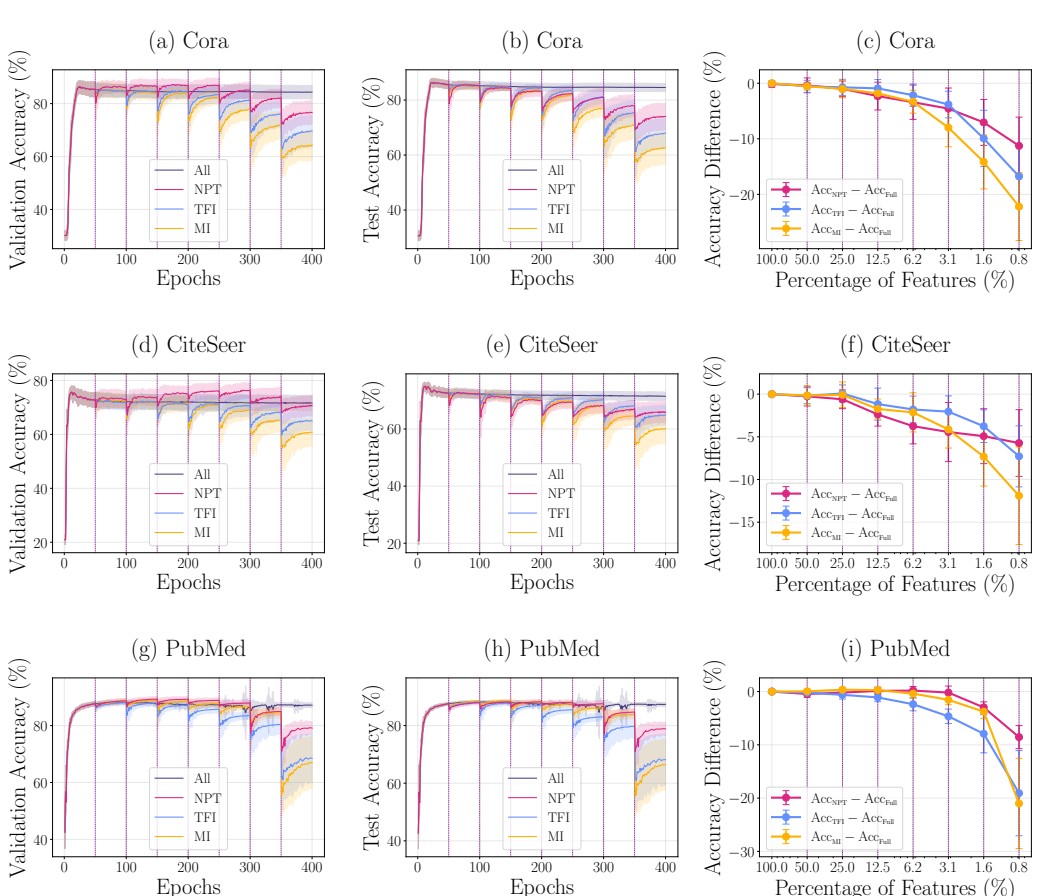

Figure 5: Node classification accuracy for homophilic datasets Cora, CiteSeer, and PubMed. (a,d,g) Validation accuracy for full, NPT, TFI, and MI. (b,e,h) Test accuracy for full, NPT, TFI, and MI. (c,f,i) Accuracy difference for full, NPT, TFI, and MI.

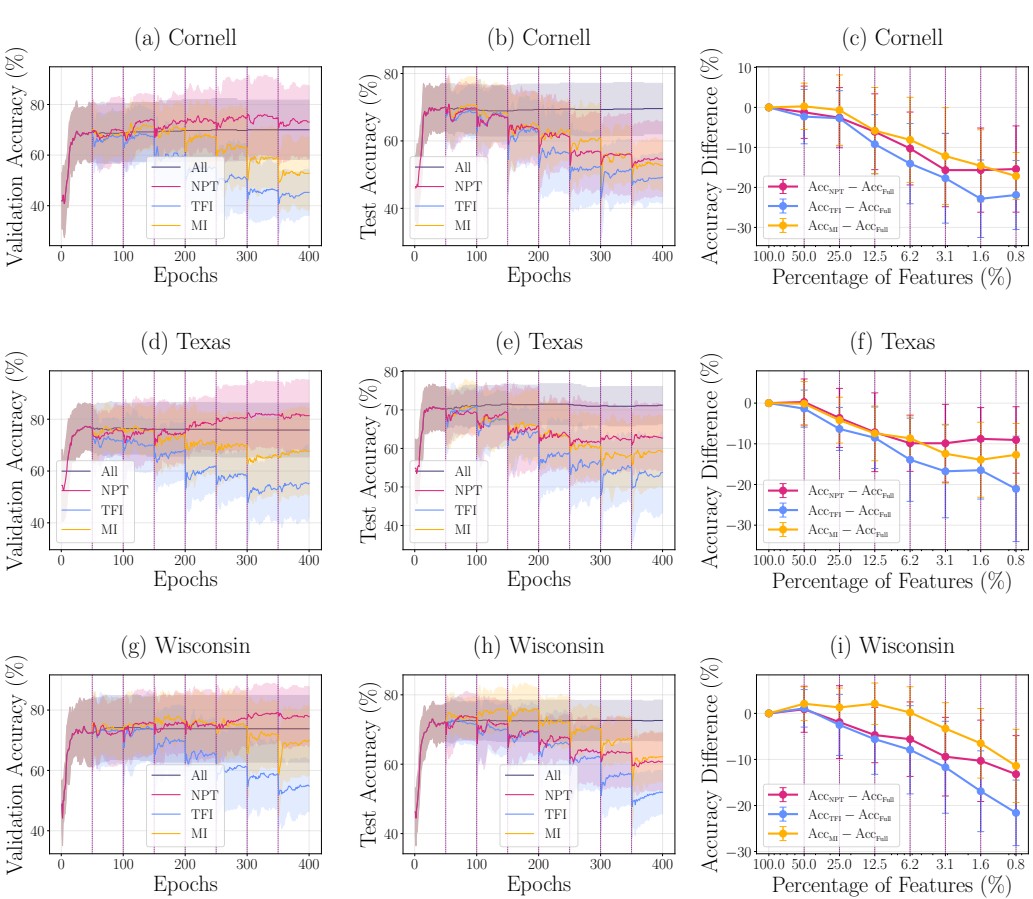

Figure 6: Node classification accuracy for heterophilic datasets Cornell, Texas, and Wisconsin. (a,d,g) Validation accuracy for full, NPT, TFI, and MI. (b,e,h) Test accuracy for full, NPT, TFI, and MI. (c,f,i) Accuracy difference for full, NPT, TFI, and MI.

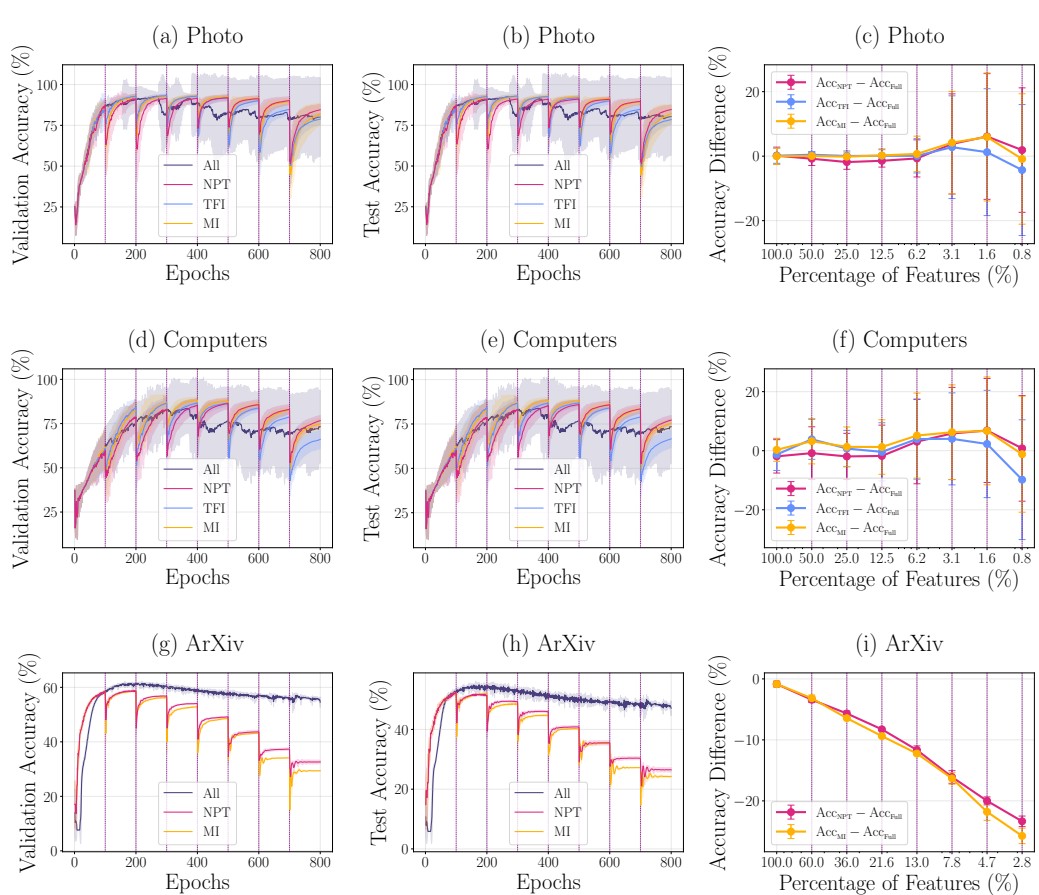

Figure 7: Node classification accuracy for larger-scale datasets Photo, Computers, and ArXiv. (a,d,g) Validation accuracy for full, NPT, TFI, and MI. (b,e,h) Test accuracy for full, NPT, TFI, and MI. (c,f,i) Accuracy difference for full, NPT, TFI, and MI.

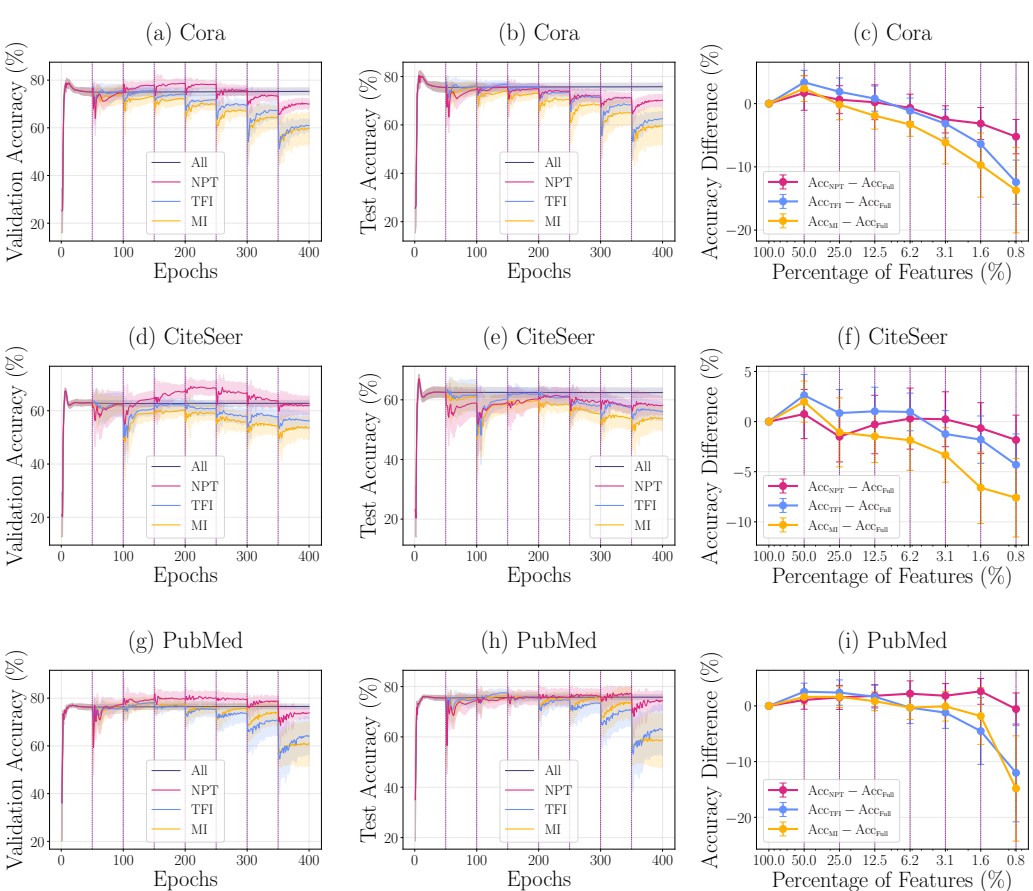

Figure 8: Node classification accuracy for homophilic datasets Cora, CiteSeer, and PubMed. Train, validation, and test node subsets are selected via canonical splits. (a,d,g) Validation accuracy for full, NPT, TFI, and MI. (b,e,h) Test accuracy for full, NPT, TFI, and MI. (c,f,i) Accuracy difference for full, NPT, TFI, and MI.

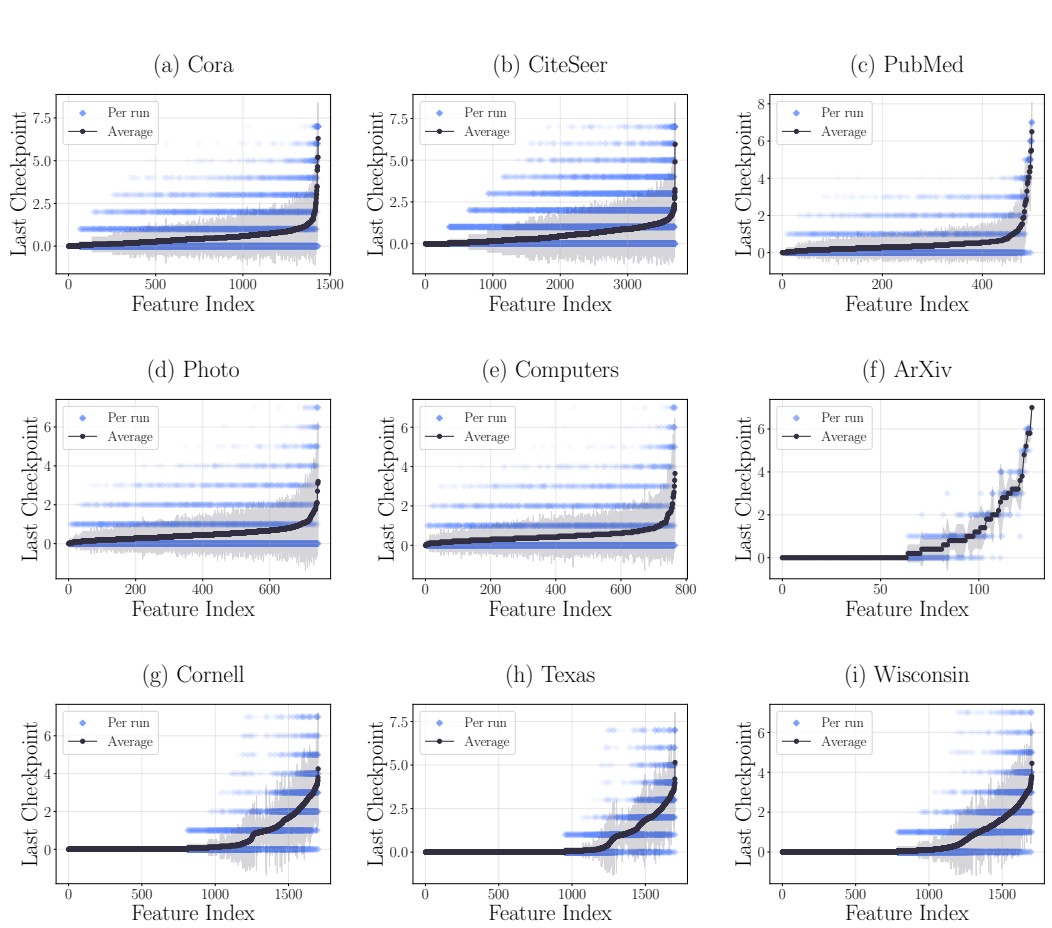

Figure 9: Last checkpoint kept per feature for various datasets. Plots (a) through (i) are presented in the following order: (a) Cora, (b) CiteSeer, (c) PubMed, (d) Photo, (e) Computers, (f) ArXiv, (g) Cornell, (h) Texas, (i) Wisconsin.

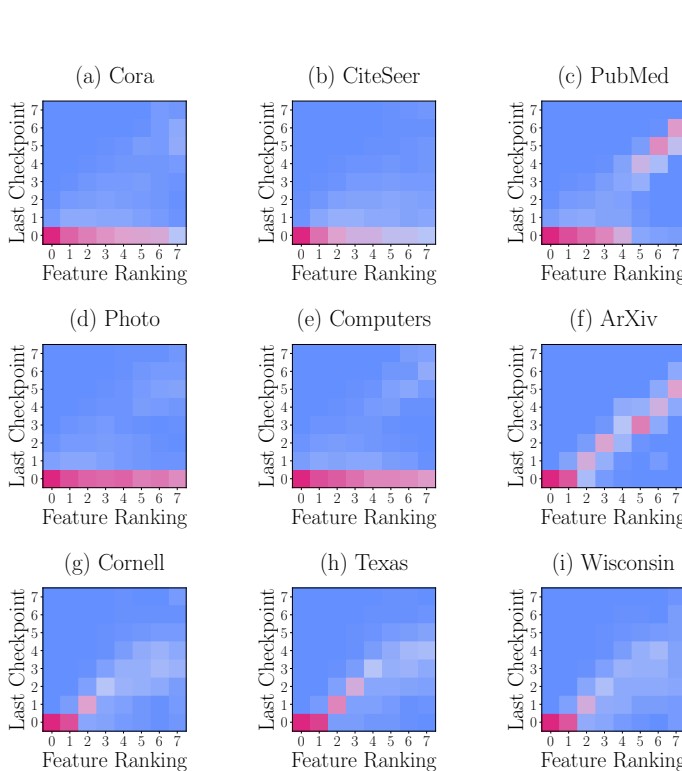

Figure 10: Last checkpoint kept per feature ranking for various datasets. Feature ranking for each feature corresponds to the most common last checkpoint the feature is kept before being dropped across independent trials. Each entry of a heatmap denotes the average last checkpoint across all features in the same ranking. Plots (a) through (i) are presented in the following order: (a) Cora, (b) CiteSeer, (c) PubMed, (d) Photo, (e) Computers, (f) ArXiv, (g) Cornell, (h) Texas, (i) Wisconsin.

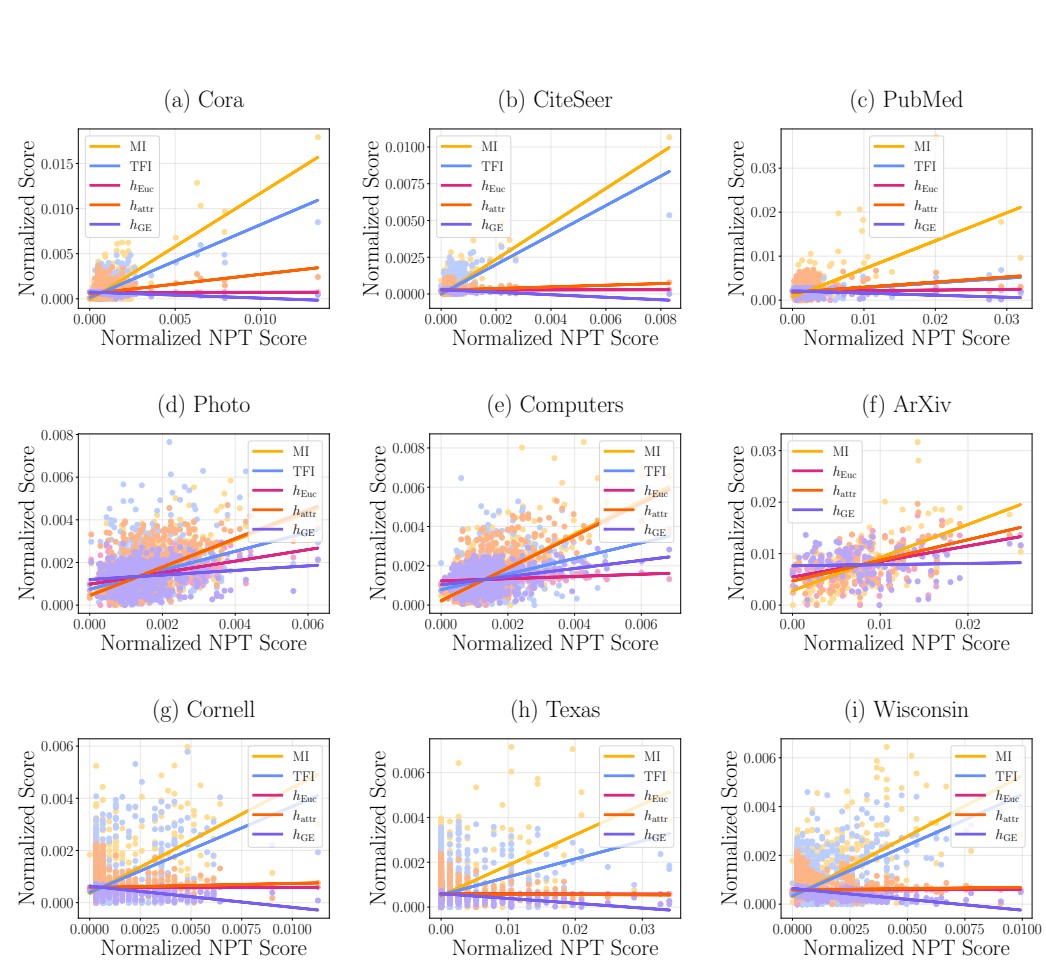

Figure 11: Plot of normalized importance scores per baseline versus normalized NPT scores. Plots (a) through (i) are presented in the following order: (a) Cora, (b) CiteSeer, (c) PubMed, (d) Photo, (e) Computers, (f) ArXiv, (g) Cornell, (h) Texas, (i) Wisconsin.

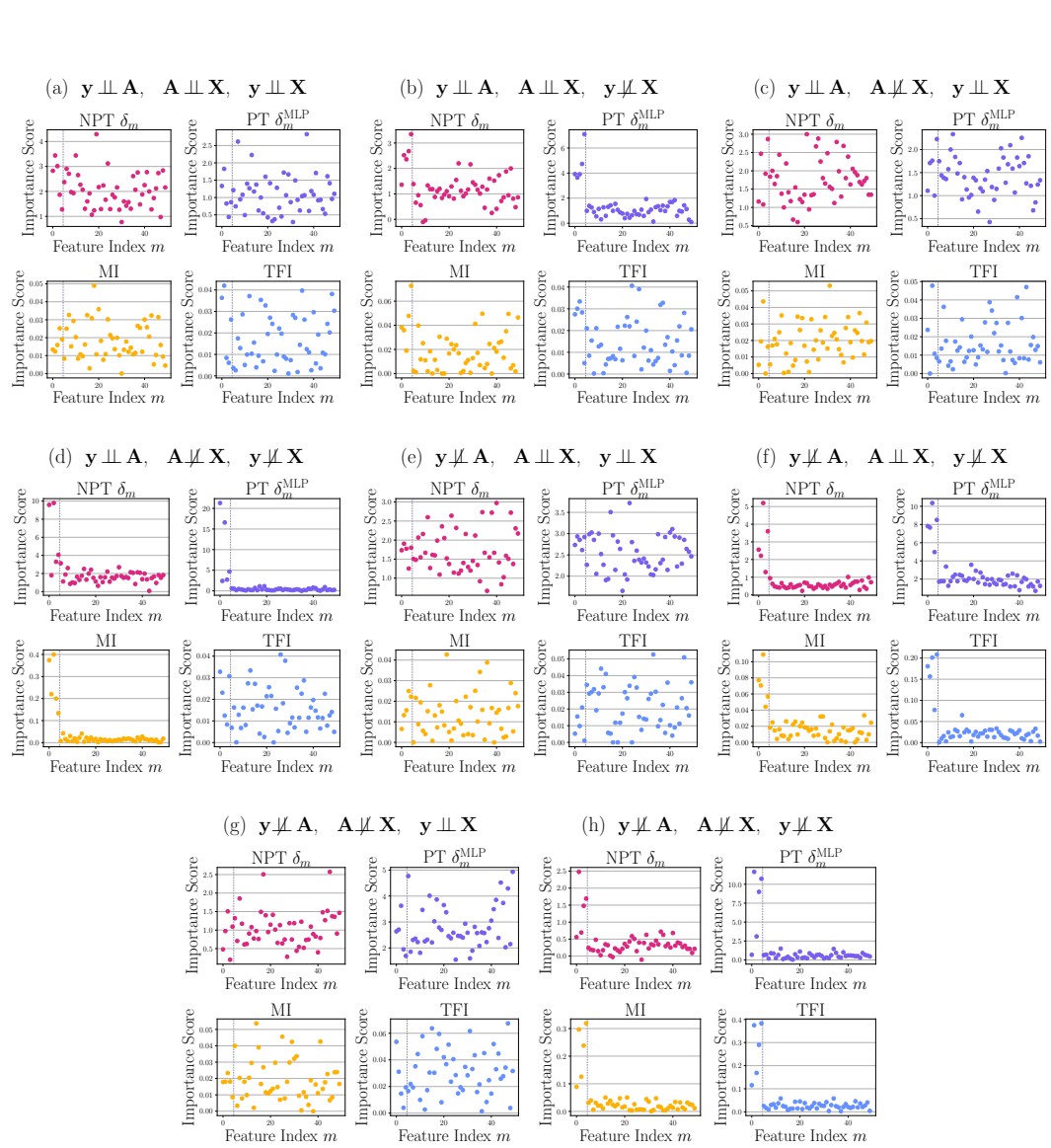

Figure 12: Feature importance scores for NPT, PT (permutation testing via MLP), TFI, and MI for synthetic datasets. Analogous to $\delta_m$ for NPT scores, $\delta_m^{\text{MLP}}$ denotes PT scores. (a) Graph, labels, and features are all independent. (b) Labels and features are correlated, but both are independent of graph. (c) Graph and features are correlated, but both are independent of labels. (d) Features are correlated with graph and labels, but labels and graph are independent. (e) Graph and labels are correlated, but both are independent of features. (f) Labels are correlated with graph and features, but graph and features are independent. (g) Graph is correlated with labels and features, but labels and features are independent. (h) Graph, labels, and features are all pairwise correlated.

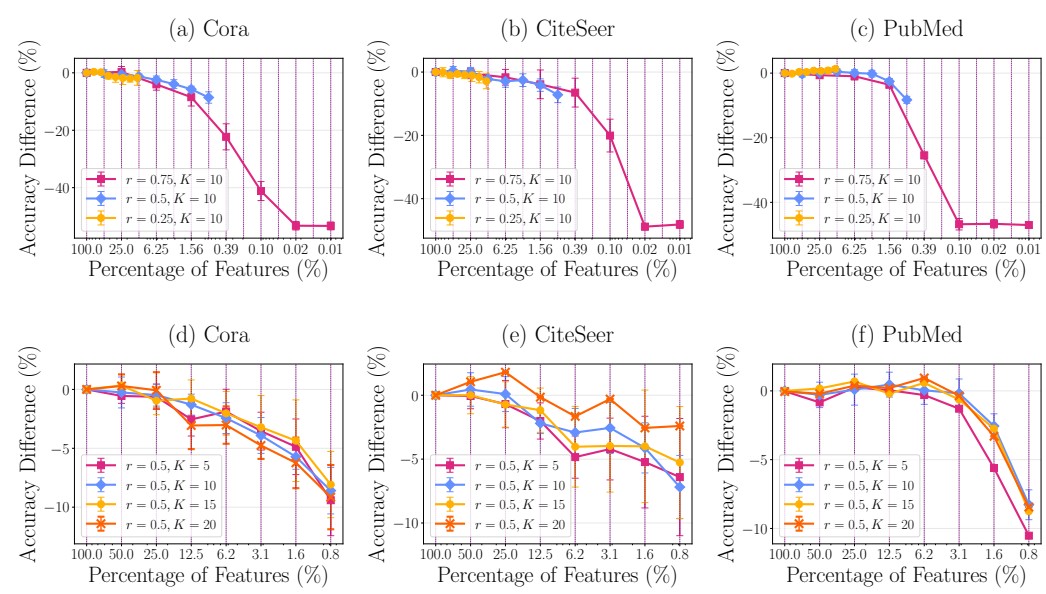

Figure 13: Hyperparameter tuning by comparing GCN performance across various dropping rates $r$ and shuffling instances $K$ for Cora, CiteSeer, and PubMed. The top row corresponds to fixing $K = 10$ while varying $r \in \{0.25, 0.5, 0.75\}$, and the bottom row denotes fixing $r = 0.5$ while varying $K \in \{5, 10, 15, 20\}$.

