# OpenReview forum: "Adaptive node feature selection for graph neural networks"
_ICLR.cc/2026/Conference — Submitted to ICLR 2026_

### Official Review · Reviewer_4vX4 · 2025-10-23

**Soundness:** 2
**Presentation:** 3
**Contribution:** 2
**Rating:** 2
**Confidence:** 4

**Summary:**

The paper proposes an adaptive node feature selection method for graph neural networks that estimates each feature’s importance during training by measuring the drop in validation performance when that feature’s values are randomly permuted, then progressively prunes low-value features via a simple routine. The authors first analyze how graph structure and node attributes jointly affect GCN behavior, deriving a bound that ties error to cross-class edges and feature similarity, and then justify permutation-based importance by showing how permuting decouples features from labels and edges in the bound, making the resulting score a proxy for influence. The approach is model- and task-agnostic, can substitute other objectives (e.g., fairness metrics) for accuracy in the importance calculation, and tracks how feature relevance evolves over time. Experiments on eight benchmarks with GCN, GIN, and TAGCN compare against mutual information, a topological feature informativeness metric, homophily scores, and random selection; results include dynamic importance heatmaps that show stable rankings over training and case studies illustrating when graph structure or attributes dominate.

**Strengths:**

The paper’s strengths are mostly conceptual and methodological. On originality, it proposes a simple, model- and task-agnostic way to score and prune node features during training via permutation tests that are explicitly tied to a GNN’s current behavior, rather than relying on graph-agnostic or architecture-specific heuristics. In quality, the work motivates the approach with analysis that relates prediction error to how labels, edges, and features interact, and then justifies permutation-based importance by showing how permuting breaks those dependencies—together with a concrete training-time procedure for adaptive pruning. Clarity is solid: the problem setup and notation are explicit, the algorithm is given in pseudocode, and the method’s “monitor as you train” design is easy to follow, including how importance is recomputed on checkpoints.

**Weaknesses:**

The empirical validation is narrow and leaves the practical value unclear: experiments cover only eight small, legacy node-classification datasets (Cora, Citeseer, PubMed; Amazon Photo/Computers; Cornell/Texas/Wisconsin) with three simple backbones (GCN, GIN, TAGConv), and five random-split runs, which is a thin basis for claims of generality; stronger evidence would add larger/modern benchmarks and architectures, more seeds, and fixed public splits. The feature-selection baselines are mostly simple filters (mutual information, topological feature informativeness, homophily metrics) plus a mask variant of the authors’ own method, omitting model-based or Shapley/attribution-style GNN selection methods that the paper cites in related work; including such baselines would better position the contribution. Results are also noisy and sometimes unconvincing: standard deviations are large on WebKB (e.g., Texas and Wisconsin), and accuracy often drops materially relative to using all features (e.g., −6–7 points on Cora after selection), so it is hard to conclude the method reliably helps. Finally, applicability is limited to node classification and the authors themselves note that correlated features can bias permutation importance; extending to link prediction/graph classification and using conditional permutation or correlation-aware corrections would strengthen the case.

**Questions:**

- Empirical setup and statistical support: the current evaluation uses five random splits with resampled masks and a fixed pruning schedule, but many results degrade or have large variance—could you expand to more seeds (e.g., 10–20), and report the training/inference overhead of the permutation procedure?
- Your evaluation uses eight small, legacy node-classification datasets with random splits—can you add modern, large-scale benchmarks (e.g., OGB, Heterophilous Node Classification) using their official splits, and report runtime/memory overhead to demonstrate scalability and practical cost?

---

> ### Author Response · Authors · 2025-11-22
> **Response to Reviewer 4vX4**
>
> We appreciate your precise summary of our contributions, and we thank you for your kind words regarding the clarity of our work.
>
> > The empirical validation is narrow and leaves the practical value unclear: ...
>
> We understand your concern regarding dataset size as some applications require very large graphs.
> However, we assert the value of using well-understood settings such as these benchmarks to evaluate a traditional statistical approach in an untested setting such as GNNs, particularly as we espouse interpretability for trustworthy graph learning. Nevertheless, we agree that your recommendations are valuable additions to our manuscript.
> We therefore added the following results.
>
> - We measure feature importance for synthetic graph data to evaluate which methods can identify relevant features in controlled settings, comparing **NPT**, **TFI**, **MI**, and **PT**, that is, permutation testing using an MLP instead of a GNN.
> - As requested, we repeated the experiments with Algorithm 1 for 20 independent runs, shown in Figures 2, 5, 6, and 7. We also repeat our experiment on Cora, CiteSeer, and PubMed using their official data splits in Figure 8.
> - Figure 4c plots the permutation time at each checkpoint, that is, every $T$ epochs for which we drop $r\%$ of features.
> - As recommended, we add a larger-scale dataset from the OGB collection, representing ArXiv publications. Results in Section 4 that pertain to ArXiv include Table 4 and Figures 3c and 4c.
>
> > The feature-selection baselines are mostly simple filters...
>
> We thank you for this comment as it pointed out an aspect of our manuscript to be addressed. We clarify that our focus is on *interpretable feature importance*, hence we are interested in comparing data-agnostic permutation testing to metrics that quantify importance based on data properties. Your comment is understandable since our experimental results mostly focus on the feature selection for GNN performance. Thus, we include further analyses exploring the importance scores obtained by all methods. We plot the scores other metrics versus those of NPT in Figures 3d and 11, and we also plot their correlation in Table 4. In all cases, we observe that NPT feature importance correlates most with the best performing metric among the baselines, despite the fact that some measure fundamentally different properties.
> This validates our approach as an effective data-agnostic method.
>
> > Results are also noisy and sometimes unconvincing: ...
>
> Regarding WebKB results, the use of GCNs, even TAGCNs with larger radii, is likely to yield noisier results since the labels are heterophilic. Indeed, we repeated the experiments in Figure 5-7 for 20 seeds as suggested by the reviewer to verify our takeaways. The noise is reduced to some extent, but our conclusions remain the same. Additionally, we seek a minimal set of the most informative features, thus we do not aim to outperform the model using all features. Instead, we wish to preserve performance as much as possible despite using a small fraction of features, such as $2\%$ or $5\%$ in Table 2. Our results in Table 2 also support that NPT is an optimal general choice, as we obtain accuracy rivaling or outperforming that of baselines, even though the best performing alternative metric varies across datasets.
>
> > Finally, applicability is limited to node classification ...
>
> We agree with the reviewer that these are interesting directions that we listed as future work in our conclusion.
> Since our theoretical results pertain to node classification, we emphasize this task, which we find is already a rich topic. Moreover, while permutation is applicable for any learning task, it will be particularly interesting to explore its effects on link prediction, thus we wish to explore it in detail in a future work. Additionally, while technically employable, we hesitate to recommend permutation testing for graph classification, as this requires permuting features for all nodes in all graphs, which may be infeasible. More efficient, parallelizable alternatives will certainly make for interesting next steps. Similarly, conditional methods are more complex than our straightforward, vanilla permutations, and we look forward to adapting it to the nontrivial setting of graph data in a longer publication.
>
> We thank the reviewer for your interest in this work. In particular, we appreciate your suggestions for additional experiments. We hope you agree that they add to the strength of our work.

---

### Official Review · Reviewer_95gK · 2025-10-29

**Soundness:** 3
**Presentation:** 2
**Contribution:** 3
**Rating:** 6
**Confidence:** 3

**Summary:**

The paper proposes a model- and task-agnostic method to identify and prune unhelpful node features for GNNs during training. The core idea is a permutation-based, performance-driven importance score: periodically permute one feature column across nodes, re-evaluate validation accuracy, and treat the accuracy drop as that feature’s importance. This directly ties selection to the end task and naturally accounts for graph–feature interactions. The authors provide theory showing how graph structure and feature distributions jointly affect GCN outputs, motivating why permutation tests reveal feature utility in graphs, and then demonstrate the method’s flexibility across homophilic/heterophilic settings and different GNN backbones.

**Strengths:**

good theoretical analysis

**Weaknesses:**

computation complexity analysis

**Questions:**

1. For the study of inter-class vs. intra-class node distinguishability, you can refer to [1].

2. Why is your define Z* being the idealized embedding? Do you mean there is no one better than it?

3. "measuring feature importance based solely on dependencies between y and X may not be sufficient" This is not a surprise, as we need to consider the synergy of feature, graph structure and label distribution together, which is well developed in [2].

4. What is the computational complexity of your proposed adaptive node feature selection in Algorithm 1?

5. What is the relationship between your proposed node feature selection and your analysis in Section 2?

6. The selected features doesn't seem to improve the performance.



[1] When do graph neural networks help with node classification? investigating the homophily principle on node distinguishability. Advances in Neural Information Processing Systems. 2024 Feb 13;36.


[2] What is missing for graph homophily? disentangling graph homophily for graph neural networks. Advances in Neural Information Processing Systems, 37, 68406-68452.

---

> ### Author Response · Authors · 2025-11-22
> **Response to Reviewer 95gK - Part 1**
>
> We are sincerely grateful for your helpful comments. We particularly appreciate how you link our contributions to relevant GNN work.
>
> **Question 1.**
>
> We thank the reviewer for the reference, which connects our feature importance work to broader GCN performance. Indeed, we developed our results to emphasize the effect of feature permutations, but you are correct that Section 2 aligns with works exploring GCN performance more generally. The reference shared by the reviewer is certainly relevant in this sense, and we have included the citation in Section 2.
>
> **Question 2.**
>
> Thank you for asking about the reason for our definition; this in an excellent question. We do not at all imply that $\mathbf{Z}^{\*}$ is the only optimal outcome for distinguishing node classes. Indeed, papers such as [1] suggested by the reviewer pioneered the distinguishability of node classes in more general scenarios, which is not our focus. Hence, we define $\mathbf{Z}^{\*}$ for the most straightforward setting to demonstrate the effect of feature permutations, which requires the more granular result in Theorem 1 as opposed to the parametric approach in [1]. We clarified our scenario by adding the following sentence to Section 2:
>
> > Note that we define $\mathbf{X}^{\*}$ as above for simplicity, representing the most straightforward relationship between informative features and labels $\mathbf{y}$; node classes containing distribution shifts can still yield informative predictions (Luan et al., 2024).
>
> **Question 3.**
>
> The reviewer is correct, our conclusion aligns with the goal of [2], which seeks to describe how all three components interact globally to impact GCN performance. The approach in [2] characterizes how this synergy affects GCN performance with respect to homophily by using a parametric model; however, this requires specific distributional assumptions on the graph data. While this leads to intuitive results on the global behavior of GCNs, we require a more granular approach for Theorems 1 and 2 to specify the effects of permutations. In particular, we provide an alternate, node-level perspective that examines GCN performance with no distributional assumptions, allowing us to identify scenarios for which permutation testing is advantageous for node classification. Thus, we emphasize the complementary nature of our approach and that of [2], which we now cite following our Theorem 2.
>
> Furthermore, our conclusions from Theorem 1 do not end with the quoted statement. We elaborate on this further in response to your question 5. We thus add the following sentence after Theorem 2:
>
> > More specifically, the bound in (4) reveals that certain compositions of features and edges may render a feature important or unimportant regardless of its relevance in the absence of the graph.
>
> We are very grateful for your comment, as we were able to enhance the strength of our claim based on your insight.
>
> **Question 4.**
>
> Thank you for the suggestion. We agree that this practical information is important for our algorithm. We therefore added a discussion on the computational complexity of our approach, written in blue in Section 3.1. Furthermore, we included an additional simulation in Figure 4c that measures the time to permute features at each checkpoint in Algorithm 1, that is, at every $n T$ for $n \in \mathbb{N}$.
>
> **Question 5.**
>
> We appreciate your curiosity regarding this connection. Section 2 seeks to characterize the effect of edges on how well node features can distinguish labels. More specifically, if $A_{ij} = 0$ for all node pairs $i,j \in [N]$ in Theorem 1, then we rely solely on feature separability to identify labels, as expected. When comparing to the graph setting $\mathbf{A} \neq \mathbf{0}$, we identify scenarios where performance improves even if features are uninformative and vice versa. This reveals certain compositions of features and edges that may render a feature important or unimportant regardless of its relevance in the absence of a graph. Thus, Section 2 sets the stage to identify when permutation testing is particularly suitable for GNN-based node classification. Then, Section 3 connects our result in Theorem 1 to that of Theorem 2, theoretically validating the use of permutations for node feature importance. Additionally, Table 1 in Section 2 validates that perturbations are effective at empirically revealing relationships among node features, graph structure, and labels, which naturally supports to our proposed approach.

---

> > ### Author Response · Authors · 2025-11-23
> > **Response to Reviewer 95gK - Part 2**
> >
> > **Question 6.**
> >
> > Your comment is indeed valid. This is related to the purpose of our work. Our goal is to train a GNN on a minimal set of the most informative features such that it best preserves the performance of a model that uses all features. In particular, we apply the classical statistics approach of permuting features to node classification, and, unlike other GNN feature importance metrics, we require no prior information about graph data or model architecture. Tables 2 and 3 and Figures 2 and 5-8 illustrate that we achieve our aim, where our NPT scores are among the top performing methods if not the most accurate one. We validate this in Figure 11 by visualizing the correlation between NPT and baseline scores across multiple datasets. We consistently observe the highest correlation between NPT and the baseline with highest accuracy in Table 2, which further supports the generality of NPT.
> >
> > We thank you again for your astute comments. Your questions relate to particularly important aspects of this work, and answering them allowed us to address those points properly.

---

### Official Review · Reviewer_TSUp · 2025-10-31

**Soundness:** 3
**Presentation:** 3
**Contribution:** 2
**Rating:** 4
**Confidence:** 4

**Summary:**

This paper introduces an adaptive node feature selection method for graph neural networks (GNNs) that dynamically identifies and removes unnecessary features during training. The approach uses permutation-based importance scores to measure feature relevance by evaluating changes in validation performance when feature values are permuted. The authors provide theoretical analysis showing how graph structure and node features jointly influence GCN performance, and demonstrate their method across multiple benchmark datasets with different graph properties (homophilic and heterophilic).

**Strengths:**

- The paper provides theoretical analysis through Theorems 1 and 2, establishing clear bounds on how graph structure and node features jointly affect GCN performance. This theoretical foundation effectively motivates why permutation-based importance scores are particularly suitable for graph-structured data, where traditional feature selection methods may fail to account for the interplay between features and graph topology.
- the proposed method is model-agnostic and task-agnostic, requiring no assumptions about graph properties or specific GNN architectures. This flexibility is demonstrated through experiments on diverse datasets, including homophilic graphs (citation networks), heterophilic graphs (webpage networks), and co-purchase graphs, using different architectures (GCN, GIN, TAGCN) for each setting.
- The algorithm's ability to progressively eliminate features during training while monitoring importance score evolution is valuable. Figure 3's heatmaps effectively show that feature importance rankings stabilize early in training, suggesting the method can identify relevant features before full convergence, potentially saving computational resources.
- Overall, the experimental evaluation is thorough, including multiple datasets with varying properties, comparison against numerous baselines (including graph-specific metrics like homophily-based scores), and analysis of the adaptive selection process over time. The inclusion of heterophilic graphs is particularly valuable as these represent challenging cases for standard GNNs.

**Weaknesses:**

- The improvements over baseline methods are often marginal. In Table 2, NPT achieves 79.19% on Cora compared to 85.83% with all features. On some datasets like Photo and Computers, even random feature selection performs competitively, raising questions about the method's practical utility when simpler approaches suffice.
- The method requires computing permutation-based scores every T epochs by evaluating the model K times per feature, which could be computationally expensive for high-dimensional datasets. The paper doesn't provide runtime comparisons or discuss the computational trade-off between feature reduction and the cost of computing importance scores.
- While the paper shows that features can be selected effectively, it doesn't provide an analysis of what types of features are being selected or removed. Understanding which features are deemed important could provide valuable insights for domain experts and help validate the method's decisions.
- The method introduces several hyperparameters (burn-in period T_burn, interval T, retention rate r, number of permutations K) but provides limited analysis of sensitivity to these choices. The fixed choice of r=0.5 (dropping half the features at each step) seems arbitrary and may not be optimal for all settings.
- The paper doesn't adequately discuss when the method fails or performs poorly. For instance, on heterophilic graphs, the performance drops are substantial, but there's limited analysis of why permutation-based importance might be less effective in these settings compared to simpler alternatives like mutual information.

**Questions:**

1. How does the computational cost of the adaptive feature selection scale with the number of features and nodes? Could you provide runtime comparisons with training using all features?
2. Can you provide an analysis of which types of features are consistently selected or removed across different datasets? This would help validate that the method is making sensible choices.
3. How sensitive is the method to the choice of hyperparameters, particularly the retention rate r? Have you experimented with adaptive scheduling of r based on the importance score distribution?
4. Why does the method perform relatively poorly on heterophilic graphs compared to mutual information? Is there a theoretical explanation for this limitation?
5. Could the permutation test be made more efficient by using importance sampling or other variance reduction techniques rather than uniform random permutations?

---

> ### Author Response · Authors · 2025-11-22
> **Response to Reviewer TSUp - Part 1**
>
> Thank you for your thorough review, both the compliments and the constructive comments. We are glad to hear that you appreciate our theoretical analysis, flexible algorithm, and experimental evaluation. What follows are responses to your comments.
>
> **Comparison to baselines.**
> The reviewer raises a valid point regarding how to determine when NPT is successful. To clarify, we do not seek to vastly outperform other feature importance metrics. These baselines are valuable in some settings if prior information is given regarding how node features, graph structure, and labels interact. However, in real-world applications, it is likely that the best metric is not known in advance. When obtaining such information is difficult or impossible, we seek *a statistically well-established approach that does not require prior knowledge*. Permutation testing is a theoretically valid and empirically effective general choice, as NPT rivals or outperforms baselines in Table 2, even when selecting a small subset of features. We select the top $r\\%$ of features for $r \in \\{ 2,5 \\}$, and discarding at least $95\\%$ will likely increase error regardless of metric. We understand that these low choices of $r\\%$ were not obvious; we added a statement to the manuscript emphasizing $r \in \\{2,5\\}$ for Table 2.
>
> The generality of NPT also holds for Photo and Computers, about which you inquired. In fact, Theorems 1 and 2 reveal why random features perform competitively for Photo and Computers: These datasets have highly homophilic labels, with the frequencies of within-class edges being $0.827$ and $0.777$, respectively, yet many of their features have lower homophily. Theorem 1 shows that, when labels are highly homophilic, random features with high variance may be preferable to features with low homophily and low variance. However, such prior information is often not known beforehand, whereas NPT is not only agnostic to graph data properties but also competitive across Tables 2 and 3 and Figures 2 and 5-8, even though the best baseline metric varies across datasets.
>
> **Computational complexity.**
> Thank you for your comment; it is indeed important to discuss the practicality of permuting during training. Please see our added discussion on computational complexity written in blue in the revised manuscript after introducing Algorithm 1. We also empirically illustrate this complexity in Figure 4c, which shows the permutation time for each checkpoint across multiple datasets.
>
> However, we agree with the reviewer that measuring the scores of every feature via uniform permutations can be computationally prohibitive for very high-dimensional node features. While we require vanilla permutations to assess the power of classical, statistics-based permutation importance for GNNs, our algorithm can accommodate more efficient metrics, such as importance sampling variance reduction techniques, as suggested by the reviewer. In fact, our theoretical results still hold as long as the alternative approach yields the same mean and covariance as node feature permutations.
>
> **Feature importance analysis.**
> We absolutely agree with the reviewer on this topic.
> Interpretability via feature importance is a primary motivation for our work, and an analysis of both the features selected and their importance scores are critical for this paper.
> We therefore include the following additional analyses:
>
> - We portray *the NPT importance of each feature across datasets* to illustrate that feature rankings follow expectations based on dataset type and our theoretical results. To this end, for every feature, Figures 3c and 9 plots the last checkpoint in which the feature is kept before being dropped, as later checkpoints indicate more important features.
> - We also compute the *correlation between each baseline metric and our NPT metric $\delta_m$*. Figures 3d and 11 plots the normalized scores of each baseline versus normalized NPT scores for all features in each dataset, which allows us to both evaluate the properties of features selected by NPT via other metrics and also identify which baseline tends to rank features most similarly to NPT. In Table 4, we compute the correlation between NPT scores **$\mathbf{\delta}$** and scores from each baseline. Crucially, NPT was consistently most correlated with the baseline that performed best in Table 2, validating NPT as a successful general approach.
> - We evaluate our method on synthetic graph data for which we control *the relationships between labels, features, and graph edges* as well as *which features are relevant*. Figure 12 visualizes the importance scores obtained from metrics **NPT**, **TFI**, **MI**, and **PT** (permutation testing using an MLP instead of a GNN).

---

> ### Author Response · Authors · 2025-11-22
> **Response to Reviewer TSUp - Part 2**
>
> **Hyperparameter selection.**
>
> Regarding the number of parameters $K$, one may choose $K$ to reduce the distance between the true and empirical expected feature permutations, which we characterize in Proposition 1 of Appendix E through a high probability bound that decays with $K$.
> We could also develop an adaptive approach that automatically chooses $T_{\rm burn}$ and $T$ based on model performance and $r$ using NPT scores during training.
> However, this can increase training time if performance converges slowly, and we cannot guarantee how many features are discarded without extra constraints.
> Thus, since the baselines consist of static measurements, we impose a fixed $r$ for a fair comparison of all methods.
> We leave the adaptive dropping rate to future research, which we believe deserves its own work for thorough exploration.
> Instead, we provide two additional simulations comparing performance for Cora, CiteSeer, and PubMed while varying either $r$ or $K$ in Figures 4a,b and 13.
>
> **Discussing method limitations.**
>
> The reviewer wisely asks for more discussion on challenging scenarios for our method, particularly for heterophilic data.
> This is a good point, as our explanation for these results were implied but not explicitly explained.
> Indeed, by Theorem 1, when labels are heterophilic, GCNs rely much more on node features to distinguish classes, and this distinguishability may be captured by MI.
> Thus, by Theorem 1 and MLP performance in Table 1, it is unsurprising that MI performs well on heterophilic datasets.
> Moreover, features in these datasets are highly homophilic, which implies a harder problem since this can promote similarity between node embeddings across classes.
> If a graph with heterophilic labels is dense, then embeddings will already be difficult to distinguish.
> In this case, permuting may not increase error greatly, and we may erroneously dub a feature less detrimental than it is.
> However, Cornell, Texas, and Wisconsin are sparse, so even with high feature homophily, the deleterious influence of the graph is reduced.
> Thus, permutation still effectively decouples features from TAGCN predictions, as we still observe competitive or superior performance for NPT versus other metrics in Tables 2 and 3 and Figures 2 and 5-8.
>
> We sincerely thank you for your thoughts and suggestions.
> We believe our additions in response to your comments have greatly improved our work.
> We hope that you find these results informative and satisfactory.

---

### Author Response · Authors · 2025-11-22
**Global comment**

We would like to thank the reviewers for their comments, questions, and suggestions.
We believe that our updates based on your reviews have greatly improved our paper.
In the revised manuscript, we denote changes in blue for convenience.
We also summarize the newly added results below.

- For our adaptive feature selection results, we include an additional dataset ArXiv from the OGB collection.
- We repeat the adaptive feature selection results in Figures 2, 5, 6, 7, and 8 for 20 independent runs rather than the previous 5.
- We include plots of the last checkpoint in which a feature is kept before being dropped in Algorithm 1. Figure 3c presents this plot for ArXiv, while all datasets are shown in Figure 9 in Appendix G. Furthermore, we group features by their most frequent last checkpoint and plot heatmaps of the average last checkpoint per feature group in Figure 10 of Appendix G.
- We plot the importance scores obtained via baselines versus our NPT importance scores. This is shown for Wisconsin in Figure 3d, while Figure 11 of Appendix G shows the same result for all datasets. Furthermore, Table 4 measures Pearson's correlation coefficient between NPT scores and baseline scores.
- We include a result in Figure 12 of Appendix G that plots the feature importance measured by our NPT metric and baseline scores for synthetic data, for which we control the relationships between graphs, labels, and features.
- We include results tuning hyperparameters $r$, which denotes the rate of dropping features, and $K$, which represents the number of feature permutations per checkpoint. Figure 4a,b present these results for PubMed, while Figure 13 in Appendix H shows the same simulation for Cora, CiteSeer, and PubMed.
- We also include a discussion on the computational complexity of our algorithm in Section 3.1, along with a simulation measuring the permutation time during training of multiple datasets in Figure 4c.
- We include a brief Proposition 1 in Appendix E that provides a high probability bound between the empirical average of feature permutations and the theoretical expected value, which serves as a guide to the number of permutations to use in our Algorithm 1.

We again thank the reviewers for their time, consideration, and thoroughness.
We hope that these updates address your comments and questions.

---

### Author Response · Authors · 2025-12-02
**Discussion Summary - Part 1**

For added convenience, we summarize the comments of the reviewers along with our corresponding answers or manuscript changes in response. We are grateful to our reviewers for helping us improve our manuscript, and we thank the area chair for considering our work.

**Reviewer TSUp.**
1. The reviewer commented on the performance of our feature importance metric (weakness 1).
	- We *clarified our purpose* to the reviewer: We seek a statistically well-established, data-agnostic approach. We also explained *performance in Table 2 and updated our explanation in the manuscript* accordingly.
2. The reviewer asked about the *computational cost* of our method along with *empirically measuring runtime* (weakness 2 and question 1). The reviewer also asked if our algorithm can be made more efficient through other sampling techniques (question 5).
	- We added a *discussion on computational complexity* in Section 3.1 after introducing our algorithm, along with a *simulation measuring permutation time* during training in Figure 4c. We also affirmed that other sampling techniques are certainly feasible, albeit not the focus of our work.
3. The reviewer requested that we *analyze the features selected* by our approach (weakness 3 and question 2).
	- We added the following results, described in detail in the updated manuscript: (1) Figures 3c, 9, and 10 depict *feature importance rankings per dataset*; (2) Table 4 and Figures 3d and 11 show the *correlation between our feature importance scores NPT and baseline metrics*; and (3) Figure 12 plots *feature importance scores for synthetic data*, where we control which features are important.
4. The reviewer asked for further discussion on limitations, particularly for heterophilic data (weakness 5 and question 4).
	- We explained why the *graph-agnostic baseline MI performs well on heterophilic data* and why *our method NPT can still rival or outperform MI*. To this end, we connected our Theorems 1 and 2 to the results in Tables 2 and 3 and Figures 2 and 5-8 in our response to the reviewer.

**Reviewer 95gK.**
1. The reviewer shared a reference relevant to our discussion in Section 2 (question 1). While our paper focuses on node feature importance, our discussion in Section 2 also relates to GCN performance in general, hence the suggested reference.
	- We *added the relevant citation* to our updated manuscript.
2. The reviewer requested clarification about *our "idealized" node embeddings* $\mathbf{Z}^{*}$ (question 2).
	- We explained that *our choice of* $\mathbf{Z}^{*}$ *is for simplicity*. *We added a statement to clarify this* in the manuscript.
3. The reviewer related our result in Theorem 1 to another suggested reference, which discusses how homophily with respect to labels, features, and edges affect GCN performance.
	- We *expanded on the different but complementary perspectives* in the suggested reference and our work. Due to its relevance, *we cited the suggested reference* after Theorem 2. Thanks to the reviewer's comment, *we added a sentence to clarify the novelty of our result*, which differs from the conclusion in the suggested reference.
4. The reviewer asked for the *computational complexity of our algorithm* (question 3).
	- Please see point 2 in the summary of our response to Reviewer TSUp.
5. The reviewer asked for the *relationship between Section 2 and our proposed node feature selection algorithm* (question 4).
	- We *walked through our discussion in Section 2*, how it pertains to node feature importance, and how it naturally leads to our algorithm in Section 3.
6. Similar to Reviewer TSUp, the reviewer commented on the performance of our approach (question 5).
	- We again *clarified the purpose of our proposed metric and algorithm*, as in point 1 in the summary of our response to Reviewer TSUp.

---

> ### Author Response · Authors · 2025-12-02
> **Discussion Summary - Part 2**
>
> **Reviewer 4vX4.**
>
> 1. The reviewer offered multiple suggestions to strengthen our empirical results (sentence 1 of weaknesses and questions 1 and 2).
> 	- The reviewer suggested considering larger graphs, such as an OGB dataset.
> 		- We *emphasized the value in using well-known datasets* for evaluating a novel, statistics-based method, but we also *implemented our algorithm on the larger-scale OGB ArXiv dataset*, as requested.
> 	- The reviewer asked for *more independent trials* for evaluating our algorithm, as well as using *official data splits*.
> 		- We repeated the experiments with Algorithm 1 for *20 independent trials*, shown in Figures 2, 5, 6, and 7. We also repeated this for Cora, CiteSeer, and PubMed *using their official splits* in Figure 8.
> 	- The reviewer requested that we *measure runtime*.
> 		- Figure 4c plots the measured permutation time of our proposed algorithm.
> 	- The reviewer's comment was based on *validating our claims of generality*.
> 		- To further assuage the reviewer's concern, we *measure feature importance using synthetic graph data* to verify that our metric identifies relevant features.
> 2. The reviewer asked about Shapley- or model-based node feature selection methods (sentence 2 of weaknesses and question).
> 	- We note that *Shapley values are infamously computationally intensive* and thus unrealistic for our algorithm. We clarified that *our focus is on interpretable node feature importance*, so we do not consider the approaches that do not return such scores. To stress our emphasis on importance scores, *we shared results on feature importance analysis* in Figures 3d and 11 and Table 4.
> 3. Similar to Reviewer TSUp, the reviewer commented on *our observed performance gaps, particularly for heterophilic data* (sentence 3 of weaknesses and question 1).
> 	- As noted above, we *repeat simulations for 20 independent trials*. Since we observed similar outcomes, we *explained both the noisy nature of GCNs for heterophilic data* as well as *reiterating our purpose of identifying a data-agnostic, general metric*.
> 4. The reviewer noted that *implementing our plans for future work would strengthen the paper* (sentence 4 of weaknesses).
> 	- We agreed that *these directions are particularly interesting* and hence deserving of their own publications.

---

### Meta-Review · Area_Chair_ZSZk · 2026-01-06

**Summary:**

The paper proposes a model and task agnostic adaptive node feature selection method for GNNs that, during training, assigns each feature a permutation-based importance score and progressively prunes low-importance features. The authors provide theoretical analysis showing how graph structure and node features jointly influence GCN performance to motivate why permutation tests capture graph–feature interactions more faithfully than graph-agnostic filters. The authors evaluated the proposed method across homophilic and heterophilic datasets.

Most of the concerns of the reviewers are still outstanding.

**Reviewer Concerns:**

Most of the concerns of the reviewers are still outstanding, here is the list:

1/ The improvements over baseline methods are often marginal. (TSUp) The proposed method delivers no material gains in either accuracy or efficiency over baseline methods.

2/ The method could be computationally expensive for high-dimensional datasets. (TSUp)

3/ The method introduces several hyperparameters but provides limited analysis of sensitivity to these choices. (TSUp)

4/ Experiments cover only eight small, legacy node-classification datasets. Experiments on larger datasets are required. (4vX4) In the rebuttal, only ogbn-arxiv was added which is not enough.

5/ Model-based or Shapley/attribution-style GNN selection methods should be added as baselines. (4vX4)

6/ Applicability is limited to node classification. (4vX4)

**Reviewer Scores:**

Two reviewers gave negative scores (2 and 4) and one reviewer gave a positive score (6). I don't think the reviewers would change their scores.

---

### Decision · Program_Chairs · 2026-01-26

Reject